# IMPROVED MUTUAL INFORMATION ESTIMATION

## ABSTRACT

We propose a new variational lower bound on the KL divergence and show that
the Mutual Information (MI) can be estimated by maximizing this bound using a
witness function on a hypothesis function class and an auxiliary scalar variable. If
the function class is in a Reproducing Kernel Hilbert Space (RKHS), this leads to
a jointly convex problem. We analyze the bound by deriving its dual formulation
and show its connection to a likelihood ratio estimation problem. We show that the
auxiliary variable introduced in our variational form plays the role of a Lagrange
multiplier that enforces a normalization constraint on the likelihood ratio. By
extending the function space to neural networks, we propose an efficient neural MI
estimator, and validate its performance on synthetic examples, showing advantage
over the existing baselines. We then demonstrate the strength of our estimator in
large-scale self-supervised representation learning through MI maximization.

## 1 INTRODUCTION

Mutual information (MI) is an ubiquitous measure of dependency between a pair of random variables,
and is one of the corner stones of information theory. In machine learning, the information maxi-
mization principle for learning representation from unlabeled data through self-supervision (Bell &
Sejnowski, 1995) motivated the development of many MI estimators and applications (Hjelm et al.,
2019; Noroozi & Favaro, 2016; Kolesnikov et al., 2019; Doersch et al., 2015; van den Oord et al.,
2018b; Hu et al., 2017). The information bottleneck (Tishby et al., 1999; Kolchinsky et al., 2017) is
another principle that triggered recent interest in mutual information estimation. MI is also used to
understand the information flow in neural networks, in learning clusters (Krause et al., 2010) and in
regularizing the training of Generative Adversarial Networks (GANs) (Chen et al., 2016).

In many of those machine learning applications and other scientific fields, one has to estimate MI
given samples from the joint distribution of high dimensional random variables. This is a challenging
problem and many methods have been devised to address it. Since MI is defined as the Kullback-
Leibler (KL) divergence between the joint distribution and the product of marginals, one can leverage
non parametric estimators of f-divergences (Nguyen et al., 2008; Nowozin et al., 2016; Sriperumbudur
et al., 2009). Specifically of interest to us is the Donsker-Varadhan (DV) representation of the KL
divergence (Donsker & Varadhan, 1976) that was used recently with neural networks estimators
(Belghazi et al., 2018; Poole et al., 2019). Other approaches to estimating the MI are through finding
lower bounds using variational Bayesian methods (Alemi et al., 2016; 2017; Barber & Agakov, 2003;
Blei et al., 2017), as well as through geometric methods like binning (Kraskov et al., 2004).

In this paper we propose a new estimator of MI that can be used in direct MI maximization or as
a regularizer, thanks to its unbiased gradients. Our starting point is the **DV** lower bound of the KL
divergence that we represent equivalently via a joint optimization that we call $\boldsymbol{\eta}$-**DV** on a witness
function $f$ and an auxiliary variable $\eta$ in Section 2. In Section 3, we show that when the witness
function $f$ is learned in a Reproducing Kernel Hilbert Space (RKHS) the $\boldsymbol{\eta}$-**DV** problem is jointly
convex in both $f$ and $\eta$. The dual of this problem sheds the light on this estimator as a constrained
ratio estimation where $\eta$ plays the role of a Lagrange multiplier that ensures proper normalization of
the likelihood ratio. We also show how the witness function can be estimated as a neural network akin
to Belghazi et al. (2018). We specify our estimator for MI in Section 4, and show how it compares
to alternatives in the literature (Nguyen et al., 2008; Belghazi et al., 2018; Poole et al., 2019). The
experiments are presented in Section 5. On synthetic data, we validate our estimators by estimating
MI on Gaussian variables and by regularizing GAN training as in Chen et al. (2016). On real data,

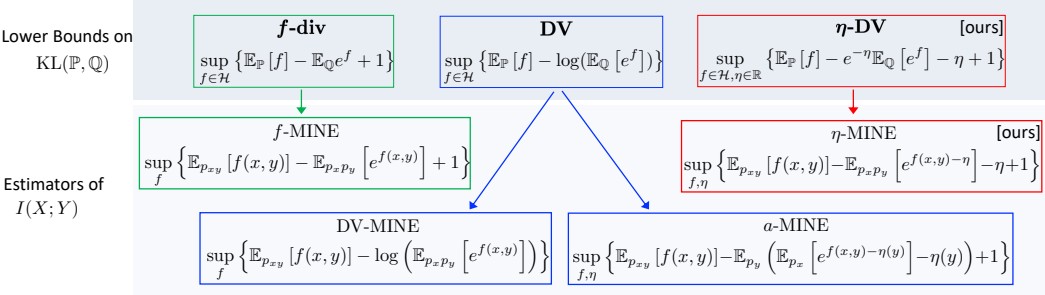

**Figure 1:** Overview of the paper. Top row shows several KL divergence lower bounds for two probability distributions $\mathbb{P}$ and $\mathbb{Q}$. By substituting $\mathbb{P} = p_{xy}$, $\mathbb{Q} = p_x p_y$, and defining $f$ as a neural network, we obtain corresponding MI estimators. $\boldsymbol{\eta}$-**DV** and $\eta$-MINE are the proposed bound and its derived estimator. Further details are provided in the text.

we explore our estimator in deep MI maximization for learning representation from unlabeled data. Figure 1 shows an overview of all the bounds and related MI estimators discussed in this paper.

## 2   LOWER BOUNDS ON KL DIVERGENCES AND MUTUAL INFORMATION

Consider two probability distributions $\mathbb{P}$ and $\mathbb{Q}$, where $\mathbb{P}$ is absolutely continuous w.r.t. $\mathbb{Q}$. Let $p$ and $q$ be their respective densities defined on $\mathcal{X} \subset \mathbb{R}^d$. Their KL divergence is defined as

$$\mathrm{KL}(\mathbb{P}, \mathbb{Q}) = \mathbb{E}_{x \sim \mathbb{P}} \log\left(\frac{p(x)}{q(x)}\right) = \int p(x) \log\left(\frac{p(x)}{q(x)}\right) dx.$$

We are interested in the MI between two random variables $X, Y$ where $X$ is defined on $\mathcal{X} \subset \mathbb{R}^{d_x}$, and $Y$ on $\mathcal{Y} \subset \mathbb{R}^{d_y}$. Let $p_{xy}$ be their joint densities and $p_x, p_y$ the marginals of $X$ and $Y$ respectively. The MI is defined as follows:

$$I(X;Y) = \mathrm{KL}(p_{xy}, p_x p_y), \tag{1}$$

which is the KL divergence between the joint density and the product of marginals. Non-parametric estimation of MI from samples is an important problem in science and machine learning. In what follows, we review variational lower bounds on KL to enable such estimation.

**Variational Characterization of KL divergence**. Let $\mathcal{H}$ be any function space mapping $\mathcal{X}$ to $\mathbb{R}$. The first variational characterization of the KL divergence goes back to Donsker & Varadhan (1976):

$$\mathrm{KL}(\mathbb{P}, \mathbb{Q}) \geq D_{\mathrm{DV}}^{\mathcal{H}}(\mathbb{P}, \mathbb{Q}) = \sup\{\mathbb{E}_{x \sim \mathbb{P}} f(x) - \log(\mathbb{E}_{x \sim \mathbb{Q}} e^{f(x)}) : f \in \mathcal{H}\}, \tag{2}$$

where the equality holds if and only if (iif) $f^* = \log(p/q) \in \mathcal{H}$. We refer to this bound as the **_DV_ bound**.

The second variational representation was introduced in Nguyen et al. (2008); Nowozin et al. (2016) and derived through convex duality to be finally stated as follows:

$$\mathrm{KL}(\mathbb{P}, \mathbb{Q}) \geq D_f^{\mathcal{H}}(\mathbb{P}, \mathbb{Q}) = 1 + \sup\{\mathbb{E}_{x \sim \mathbb{P}} f(x) - \mathbb{E}_{x \sim \mathbb{Q}} e^{f(x)} : f \in \mathcal{H}\}, \tag{3}$$

with equality iif $f^* = \log(p/q) \in \mathcal{H}$. We call this bound the **_f_-div** bound (as in $f$-divergence).

From Eq. 2 and Eq. 3 we see that the variational bounds are attempting to estimate the log-likelihood ratio $f^* = \log(p/q)$, and the tightness of the bound depends on the representation power of $\mathcal{H}$. In order to compare these two lower bounds, observe that $\log(t) \leq t - 1, t > 0$. Therefore $\log\left(\mathbb{E}_{x \sim \mathbb{Q}} e^{f(x)}\right) \leq \mathbb{E}_{x \sim \mathbb{Q}} e^{f(x)} - 1$ which means that for any function space $\mathcal{H}$ we have:

$$\mathrm{KL}(\mathbb{P}, \mathbb{Q}) \geq D_{\mathrm{DV}}^{\mathcal{H}}(\mathbb{P}, \mathbb{Q}) \geq D_f^{\mathcal{H}}(\mathbb{P}, \mathbb{Q}), \tag{4}$$

from which we conclude that the **DV** bound is tighter than the **_f_-div** bound.

Now, given samples $\{x_i, i = 1 \ldots N, x_i \sim \mathbb{P}\}$, $\{y_i, i = 1 \ldots N, y_i \sim \mathbb{Q}\}$, estimating the KL divergence can be done by computing the variational bound from Monte-Carlo simulation. Specifically,

for the **DV** bound we have the following estimator:

$$\widehat{D}_{\mathrm{DV}}^{\mathscr{H}}(\mathbb{P}, \mathbb{Q}) = \sup \left\{ \frac{1}{N} \sum_{i=1}^{N} f(x_i) - \log \left( \frac{1}{N} \sum_{i=1}^{N} e^{f(y_i)} \right) : f \in \mathscr{H} \right\}. \tag{5}$$

Note that the Mutual Information Neural Estimator (MINE) (Belghazi et al., 2018) considered the above formulation with the hypothesis class $\mathscr{H}$ being a neural network. For the **$f$-div** bound we have similarly the following estimator:

$$\widehat{D}_{f}^{\mathscr{H}}(\mathbb{P}, \mathbb{Q}) = 1 + \sup \left\{ \frac{1}{N} \sum_{i=1}^{N} f(x_i) - \frac{1}{N} \sum_{i=1}^{N} e^{f(y_i)} : f \in \mathscr{H} \right\}, \tag{6}$$

for which Nguyen et al. (2008) introduced and studied a convex estimator with $\mathscr{H}$ being an RKHS.

While **DV** bound is tighter than the **$f$-div** bound, in order to learn the function $f$ using stochastic gradient optimization, **$f$-div** is a better fit because the cost function is linear in the expectation, whereas in the **DV** bound we have a $\log$ non-linearity being applied to the expectation. This non-linearity introduces biases in the mini-batch estimation of the cost function as noted in Belghazi et al. (2018). In the following, we show how to alleviate this problem and remove the non-linearity at the price of an additional auxiliary variable that will enable better optimization for the **DV** bound.

**An $\eta$-trick for the DV Bound.** We start with the following elementary Lemma, that gives a variational characterization of the log. All proofs are given in the Appendix A.

**Lemma 1.** *Let $x > 0$, we have:* $\log(x) = \min_{\eta} e^{-\eta} x + \eta - 1$.

Using Lemma 1 we can now linearize the log in the **DV** bound.

**Lemma 2** ($\eta-$Donsker-Varadhan). *Let $\mathscr{H}$ be any function space mapping $\mathcal{X}$ to $\mathbb{R}$:*

$$\mathrm{KL}(\mathbb{P}, \mathbb{Q}) \geq D_{\eta\text{-}DV}^{\mathscr{H}}(\mathbb{P}, \mathbb{Q}) = -\inf\{L(f, \eta) : f \in \mathscr{H}, \eta \in \mathbb{R}\} \tag{7}$$

$$L(f, \eta) = e^{-\eta} \mathbb{E}_{x \sim \mathbb{Q}} e^{f(x)} - \mathbb{E}_{x \sim \mathbb{P}} f(x) + \eta - 1, \tag{8}$$

*We refer to this bound as **$\eta$-DV** bound.*

Note that for $\eta = 0$ we recover the **$f$-div** bound. Using Lemma 2, we can now rewrite the estimator for the **DV** bound in Eq. 5 as follows:

$$\widehat{D}_{\mathrm{DV}}^{\mathscr{H}}(\mathbb{P}, \mathbb{Q}) = -\inf_{f, \eta} \hat{L}(f, \eta), \quad \text{where } \hat{L}(f, \eta) = e^{-\eta} \frac{1}{N} \sum_{i=1}^{N} e^{f(y_i)} - \sum_{i=1}^{N} f(x_i) + \eta - 1, \tag{9}$$

enabling unbiased stochastic gradient optimization of the function $f$. We note that similar variational tricks of non-linearities have been devised for $g(\eta) = \sqrt{\eta}$ in Argyriou et al. (2008); Bach et al. (2011).

## 3 $\eta$-DV BOUND, CONVEXITY, MMD RATIO ESTIMATES AND LAGRANGIANS

We established that the **DV** bound is tighter than the **$f$-div** and derived **$\eta$-DV** as an alternative to **DV**, enabling unbiased stochastic optimization. However, the role of $\eta$ in improving the lower bound remains unclear. To highlight its role, we consider the **$\eta$-DV** bound from Eq. 7, and restrict the hypothesis function class to an RKHS, similarly to Nguyen et al. (2008). Working in the dual, we can compare **$\eta$-DV** estimator to the **$f$-div** estimator and further understand this hierarchy of lower bounds. See Figure 2 for an illustration.

For simplicity, and without loss of generality, we let RKHS be a finite dimensional feature map, i.e., $\mathscr{H} = \{f | f(x) = \langle w, \Phi(x) \rangle, \Phi : \mathcal{X} \to \mathbb{R}^m, w \in \mathbb{R}^m\}$. Now for $f \in \mathscr{H}$, the loss given in Eq. 8 for **$\eta$-DV** can be rewritten as follows:

$$L(f, \eta) \triangleq L(w, \eta) = e^{-\eta} \mathbb{E}_{x \sim \mathbb{Q}} e^{\langle w, \Phi(x) \rangle} - \langle w, \mathbb{E}_{x \sim \mathbb{P}} \Phi(x) \rangle + \eta - 1. \tag{10}$$

Following Nguyen et al. (2008), we consider the following regularized loss: $\mathscr{L}(w, \eta) = L(w, \eta) + \Omega(w)$, and the corresponding sample-based formulation $\hat{\mathscr{L}}(w, \eta) = \hat{L}(w, \eta) + \Omega(w)$, where $\Omega(w)$ is a convex regularizer, e.g., $\Omega(w) = \frac{\lambda}{2} \|w\|_2^2$. The **$\eta$-DV** primal problem can now be defined as:

$$\boldsymbol{\eta}\text{-}\mathbf{DV}\text{–P} : \min_{w, \eta} L(w, \eta) + \Omega(w). \tag{11}$$

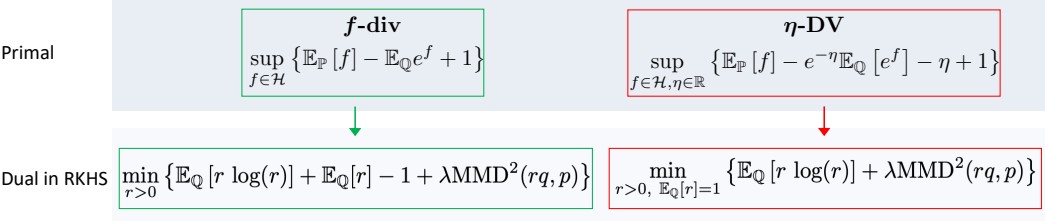

**Figure 2:** Comparison of dual formations of $\boldsymbol{\eta}$-**DV** and $\boldsymbol{f}$-**div** in RKHS. Here $r$ is an estimator for density ratio $p/q$ and $\text{MMD}_\Phi(\mathbb{P}, \mathbb{Q}) = \|\mathbb{E}_{x \sim \mathbb{P}} \Phi(x) - \mathbb{E}_{y \sim \mathbb{Q}} \Phi(y)\|$. As can be seen, both duals have a relative entropy term, but the $\boldsymbol{f}$-**div** bound does not impose the normalization constraint on the ratio, which biases the estimate, while $\boldsymbol{\eta}$-**DV** uses $\eta$ as the Lagrangian multiplier to impose the constraint $\mathbb{E}_\mathbb{Q}[r] = 1$ and ensure density normalization.

In the following, we show that $\mathscr{L}(w, \eta)$ is jointly convex in $(w, \eta)$ and derive its dual, which will shed light on the nature of the likelihood ratio estimate as well as the role of $\eta$.

**Convex Estimate in RKHS.** In Lemma 3 we first establish the convexity of the $\boldsymbol{\eta}$-**DV** loss function.

**Lemma 3.** $\mathscr{L}$ and $\hat{\mathscr{L}}$ are jointly convex in $w$ and $\eta$.

$\hat{\mathscr{L}}_\varepsilon(u, \eta) = \hat{L}(u, \eta) + \Omega(u) + \varepsilon e^{-\eta}$. Our KL estimate is therefore given as $\widehat{D}_{\text{DV}}^{\mathscr{H}}(\mathbb{P}, \mathbb{Q}) = -\hat{L}(w^*, \eta^*)$, where $(w^*, \eta^*) = \arg\min_{(u, \eta)} \hat{\mathscr{L}}_\varepsilon(u, \eta)$.

**$\boldsymbol{\eta}$-DV Dual is a Constrained Likelihood Ratio Estimation.** Another angle for estimating the KL divergence is through likelihood ratio estimation (Mohamed & Lakshminarayanan, 2016). Given a ratio estimate $\hat{r}$ of $p/q$, the KL divergence can be computed as $\mathbb{E}_\mathbb{Q} \hat{r} \log(\hat{r})$, which can be easily estimated using samples from $\mathbb{Q}$.

In Theorem 1, proven in Appendix A, we show that the dual problem (denoted D), corresponding to the primal minimization problem (denoted P) of $\boldsymbol{\eta}$-**DV**, reduces to a constrained likelihood ratio estimation problem (denoted C).

**Theorem 1.** *Let $\Omega^*(.)$ be the Fenchel conjugate of $\Omega(.)$. The $\boldsymbol{\eta}$-DV bound restricted to an RKHS, amounts to the following regularized convex minimization: $P = -(\min_{w, \eta} L(w, \eta) + \Omega(w))$, that has the following dual form:*

$$D = \min_{r>0} \max_\eta \int_\mathcal{X} r(y) \log(r(y)) q(y) dy + (\eta - 1) \left( \int_\mathcal{X} r(y) q(y) dy - 1 \right) + \Omega^*(\Delta(r)),$$

*where $\Delta(r) = \int_\mathcal{X} \Phi(x) p(x) dx - \int_\mathcal{X} r(y) q(y) \Phi(y) dy$.*

*Noticing that $\eta - 1$ plays the role of a Lagrangian multiplier, this is equivalent to the following likelihood ratio estimation problem:*

$$C = \min_{r>0} \int_\mathcal{X} r(y) \log(r(y)) q(y) dy + \Omega^* \left( \int_\mathcal{X} \Phi(x) p(x) dx - \int_\mathcal{X} r(y) q(y) \Phi(y) dy \right)$$
$$\text{subject to: } \int_\mathcal{X} r(y) q(y) dy = 1.$$

*Therefore, we have $P = D = C$. Let $(w^*, \eta^*)$ be an optimizer of P. Let $r^*$ be an optimizer of D, then the KL estimate is given by: $D_{DV}^{\mathscr{H}}(\mathbb{P}, \mathbb{Q}) = \int_\mathcal{X} r^*(y) \log(r^*(y)) q(y) dy = L(w^*, \eta^*)$.*

The regularizer $\Omega(w) = \frac{\lambda}{2} \|w\|^2$ can now be given the following interpretation based on the results of Theorem 1. Recall the definition of the MMD (Gretton et al., 2012) between distributions using an RKHS: $\text{MMD}_\Phi(\mathbb{P}, \mathbb{Q}) = \|\mathbb{E}_{x \sim \mathbb{P}} \Phi(x) - \mathbb{E}_{y \sim \mathbb{Q}} \Phi(y)\|$. Replacing the Fenchel conjugate $\Omega^*(.)$ by its expression in Theorem 1, we see that $\boldsymbol{\eta}$-**DV** is equivalent to the following dual ($\boldsymbol{\eta}$-**DV**–D)

$$\boldsymbol{\eta}\text{-}\mathbf{DV}\text{–D}: \min_{r>0} \max_\eta \int_\mathcal{X} r(y) \log(r(y)) q(y) dy + (\eta - 1) \left( \int_\mathcal{X} r(y) q(y) dy - 1 \right) + \frac{1}{2\lambda} \text{MMD}_\Phi^2(rq, p) \tag{12}$$

which can be written as a constrained ratio estimation problem ($\boldsymbol{\eta}$-**DV**–C)

$$\boldsymbol{\eta}\text{-}\mathbf{DV}\text{–C}: \min_{r>0} \int_\mathcal{X} r(y) \log(r(y)) q(y) dy + \frac{1}{2\lambda} \text{MMD}_\Phi^2(rq, p), \text{ subject to: } \int_\mathcal{X} r(y) q(y) dy = 1. \tag{13}$$

Hence, it is clear now that the $\boldsymbol{\eta}$-**DV** optimization problem is equivalent to the constrained likelihood ratio estimation problem $r$ given in Eq. 13, where the ratio is estimated using the MMD distance in the RKHS between $rq$ and $p$. It is also easy to see that $p/q$ is a feasible point and for the universal feature map $\text{MMD}_\Phi(rq, p) = 0$ iif $r = p/q$, therefore, for a universal kernel, $p/q$ is optimal and we recover the KL divergence for $r = p/q$. When comparing $\boldsymbol{\eta}$-**DV**–D from Eq. 12 and $\boldsymbol{\eta}$-**DV**–C from Eq. 13, we see that $\eta - 1$ plays the role of a Lagrangian that ensures that $rq$ is indeed a normalized distribution. In practice, we solve the primal problem (P), while the dual problem (D) and its equivalent constrained form (C) explain why this formulation estimates the KL divergence and the role of $\eta$ as a **Lagrangian** multiplier that enforces a normalization constraint. Let $r^*$ be the solution, then we have:

$$D_{\text{DV}}^{\mathscr{H}}(\mathbb{P}, \mathbb{Q}) = \int_{\mathcal{X}} r^*(y) \log\left(r^*(y)\right) q(y) dy = -L(w^*, \eta^*). \tag{14}$$

For comparison, the $\boldsymbol{f}$-**div** bound, restricted to an RKHS, is equivalent to the following ratio estimation (follows from the proof of Theorem 1 by eliminating $\max$ on $\eta$, and setting $\eta = 0$), and is consistent with the results of Nguyen et al. (2008) (see Eq. 51 therein):

$$\boldsymbol{f}\text{-}\mathbf{div} : \min_{r>0} \int_{\mathcal{X}} r(y) \log\left(r(y)\right) q(y) dy + 1 - \int_{\mathcal{X}} r(y) q(y) dy + \frac{1}{2\lambda} \text{MMD}_\Phi^2(rq, p). \tag{15}$$

Let $r^*$ be the optimum, then the KL divergence can be estimated as follows Nguyen et al. (2008):

$$D_f^{\mathscr{H}}(\mathbb{P}, \mathbb{Q}) = \int_{\mathcal{X}} r^*(y) \log\left(r^*(y)\right) q(y) dy + \left(1 - \int_{\mathcal{X}} r^*(y) q(y) dy\right). \tag{16}$$

Comparing $D_{\text{DV}}^{\mathscr{H}}(\mathbb{P}, \mathbb{Q})$ and $D_f^{\mathscr{H}}(\mathbb{P}, \mathbb{Q})$ we see that they both have a relative entropy term but the $\boldsymbol{f}$-**div** bound does not impose the normalization constraint on the ratio, which biases the estimate. We end with the following remark: In Theorem 1, if we replace the loss $L$ by its empirical counterpart $\hat{L}$ from Eq. 9, the equivalent Dual takes the following form:

$$\min_{r_i>0} \frac{1}{N} \sum_{i=1}^N r_i \log\left(r_i\right) + \Omega^*\left(\frac{1}{N} \sum_{i=1}^N \Phi(x_i) - \frac{1}{N} \sum_{i=1}^N r_i \Phi(y_i)\right), \text{ subject to: } \frac{1}{N} \sum_{i=1}^N r_i = 1,$$

and the KL estimate is given by: $\widehat{D}_{\text{DV}}^{\mathscr{H}}(\mathbb{P}, \mathbb{Q}) = \frac{1}{N} \sum_{i=1}^N r_i^* \log\left(r_i^*\right) = -\hat{L}(w^*, \eta^*)$.

**From RKHS to Neural Estimation.** One shortcoming of the RKHS approach is that the method depends on the choice of feature map $\Phi$. We propose to learn $\Phi$ as a deep neural network as in MINE (Belghazi et al., 2018). Given samples $x_i$ from $\mathbb{P}$, $y_i$ from $\mathbb{Q}$, the KL estimation problem becomes:

$$\widehat{D}_{\text{DV}}^{\text{NN}}(\mathbb{P}, \mathbb{Q}) = -\left(\min_\Phi \min_{\eta, w} e^{-\eta} \frac{1}{N} \sum_{i=1}^N e^{\langle w, \Phi(y_i)\rangle} - \sum_{i=1}^N \langle w, \Phi(x_i)\rangle + \eta - 1\right), \tag{17}$$

which can be solved using BCD on $(w, \eta, \Phi)$. Note that if $\Phi(\cdot)$ is known and fixed, then optimization problem in Eq. 17 becomes convex in $\eta$ and $w$. We refer to this bound as $\boldsymbol{\eta}$-**DV**–convex.

**What Do Neural Estimators of KL or MI Learn?** We conclude with the following observations:

1. *Ratio estimation via Feature Matching.* KL divergence estimation with variational bounds boils down to a ratio estimation using a form of MMD matching.

2. *Choice of Feature/Architecture.* The choice of the feature space or the architecture of the neural network introduces a bias in the ratio estimation; also observed in Tschannen et al. (2019).

3. *Ratio Normalization.* $\boldsymbol{\eta}$-**DV** bound introduces a normalization constraint on the ratio, that ensures a better consistent estimate.

4. *Impact of Regularizers.* The choice of the regularizer dictates the choice of the metric. For instance, Ozair et al. (2019) used gradient penalties as a regularizer, and this corresponds to a ratio matching in the Sobolev discrepancy sense. Please see Appendix A.1 for more details.

## 4 $\eta$-DV MUTUAL INFORMATION ESTIMATION

We now specify the KL estimators given in Section 3 for the MI estimation problem. Given a function space $\mathscr{H}$ defined on $\mathcal{X} \times \mathcal{Y}$,

$$I(X; Y) \geq I_{\text{DV}}^{\mathscr{H}}(X; Y) = \sup_{f \in \mathscr{H}} \mathbb{E}_{p_{xy}} f(x, y) - \log\left(\mathbb{E}_{p_x} \mathbb{E}_{p_y} e^{f(x,y)}\right) \geq I_{f\text{-div}}^{\mathscr{H}}(X; Y), \tag{18}$$

where $I_{f\text{-div}}^{\mathscr{H}}(X;Y) = \sup_{f \in \mathscr{H}} \mathbb{E}_{p_{xy}} f(x,y) - \mathbb{E}_{p_x} \mathbb{E}_{p_y} e^{f(x,y)}$.

Equivalently, with the $\eta$-trick we have:

$$I_{\text{DV}}^{\mathscr{H}}(X;Y) = -\inf_{f \in \mathscr{H}, \eta} e^{-\eta} \mathbb{E}_{p_x p_y} e^{f(x,y)} - \mathbb{E}_{p_{xy}} f(x,y) + \eta - 1. \tag{19}$$

Now, given iid samples from marginals $x_i \sim p_x$ and $\tilde{y}_i \sim p_y, i = 1 \dots N$, and samples from the joint $(x_i, y_i) \sim p_{xy}, i = 1 \dots N$, we can estimate the MI as follows:

$$\hat{I}_{\text{DV}}^{\mathscr{H}}(X,Y) = \hat{D}_{\text{DV}}^{\mathscr{H}}(p_{x,y}, p_x p_y) = -\inf_{f \in \mathscr{H}, \eta} e^{-\eta} \frac{1}{N} \sum_{i=1}^{N} e^{f(x_i, \tilde{y}_i)} - \frac{1}{N} \sum_{i=1}^{N} f(x_i, y_i) + \eta - 1, \tag{20}$$

when $\mathscr{H}$ is an RKHS, $\eta$ can be seen as a Lagrangian ensuring that the ratio estimation $r(x,y)$ of $\frac{p_{xy}}{p_x p_y}$ is normalized when integrated on $p_x p_y$, i.e., $\eta$ is a Lagrangian associated with the constraint $\int_{\mathcal{X} \times \mathcal{Y}} r(x,y) p_x(x) p_y(y) dx dy = 1$. In Table 1 we review other variational bounds for MI based on $\boldsymbol{f}$-div, **DV**, and $\boldsymbol{\eta}$-**DV** bounds (a-MINE from Poole et al. (2019) is discussed next).

| MI Estimator (bound) | Loss to minimize $\hat{L}$ | Constraints |
|---|---|---|
| DV-MINE (**DV**) | $\log\left(\frac{1}{N}\sum_{i=1}^{N} e^{f(x_i, \tilde{y}_i)}\right) - \frac{1}{N}\sum_{i=1}^{N} f(x_i, y_i)$ | $f$ is a DNN |
| Unbiased DV-MINE | $\frac{\frac{1}{N}\sum_{i=1}^{N} e^{f(x_i, \tilde{y}_i)}}{m} - \frac{1}{N}\sum_{i=1}^{N} f(x_i, y_i)$ | $f$ is a DNN, $m$ running avg. of $\frac{1}{N}\sum_{i=1}^{N} e^{f(x_i, \tilde{y}_i)}$ |
| f-MINE (**$\boldsymbol{f}$-div**) | $\frac{1}{N}\sum_{i=1}^{N} e^{f(x_i, \tilde{y}_i)} - \frac{1}{N}\sum_{i=1}^{N} f(x_i, y_i) - 1$ | $f$ is in RKHS (convex) or $f$ is a DNN |
| a-MINE (**DV**) | $\frac{1}{N}\sum_{i=1}^{N}\left(\frac{e^{f(x_i, \tilde{y}_i)}}{a(\tilde{y}_i)} + \log(a(\tilde{y}_i))\right) - \frac{1}{N}\sum_{i=1}^{N} f(x_i, y_i) - 1$ | $a$ is a DNN, $a > 0$ $f$ is a DNN |
| InfoNCE (-) | $\frac{1}{N}\sum_{i=1}^{N}\left(\log\frac{1}{N}\sum_{j=1}^{N} e^{f(x_i, \tilde{y}_j)} - f(x_i, y_i)\right)$ | $f$ is a DNN |
| $\eta$-MINE (ours:$\boldsymbol{\eta}$-**DV**) | $e^{-\eta}\frac{1}{N}\sum_{i=1}^{N} e^{f(x_i, \tilde{y}_i)} - \sum_{i=1}^{N} f(x_i, y_i) + \eta - 1$ | $f$ is in RKHS or $f$ is a DNN; $\eta \in \mathbb{R}$ |
| $\eta$-MINE-convex (ours:$\boldsymbol{\eta}$-**DV**) | $e^{-\eta}\frac{1}{N}\sum_{i=1}^{N} e^{\langle w, \Phi(x_i, \tilde{y}_i)\rangle} - \sum_{i=1}^{N} \langle w, \Phi(x_i, y_i)\rangle + \eta - 1$ | $\Phi(\cdot)$ is fixed $w \in \mathbb{R}^{\dim(\Phi)}, \eta \in \mathbb{R}$ |

**Table 1:** Given iid samples from marginals $x_i \sim p_x$ and $\tilde{y}_i \sim p_y$ and samples from the joint $(x_i, y_i) \sim p_{xy}$, we list some MI estimators, corresponding variational bounds, associated losses, and constraints on the function space. MI estimators include biased and unbiased DV-MINE from **DV** (Belghazi et al., 2018), f-MINE from $\boldsymbol{f}$-**div** (Nguyen et al., 2008; Nowozin et al., 2016), InfoNCE (van den Oord et al., 2018a), a-MINE from **DV** (Poole et al., 2019) and $\eta$-MINE, $\eta$-MINE-convex from $\boldsymbol{\eta}$-**DV**(ours).

**Improved Donsker-Varadhan for Mutual Information and $a$-DV Bound Discussion.** In the following Lemma we show that for the particular case of MI estimation, the **DV** bound can be made tighter:

**Lemma 4.** *For any function $g$ that maps $\mathcal{X} \times \mathcal{Y}$ to $\mathbb{R}^+$ we have:*

$$\text{KL}(p_{x,y}, p_x p_y) + \mathbb{E}_{p_x} \log(\mathbb{E}_{p_y} g(x,y)) \geq \mathbb{E}_{p_{x,y}} \log(g(x,y))$$

*Therefore: $I(X;Y) = \sup_{f \in \mathscr{G}(\mathcal{X} \times \mathcal{Y})} \mathbb{E}_{p_{x,y}} f(x,y) - \mathbb{E}_{p_y} \log(\mathbb{E}_{p_x} e^{f(x,y)})$ where $\mathscr{G}(\mathcal{X} \times \mathcal{Y})$ is the entire space of functions defined on $\mathcal{X} \times \mathcal{Y}$. For any function space $\mathscr{H}$ map $\mathcal{X} \times \mathcal{Y}$ to $\mathbb{R}$ we have:*

$$I(X;Y) \geq I_{a\text{-DV}}^{\mathscr{H}}(X;Y) := \sup_{f \in \mathcal{H}} \mathbb{E}_{p_{x,y}} f(x,y) - \mathbb{E}_{p_y} \log(\mathbb{E}_{p_x} e^{f(x,y)}).$$

Using Jensen's inequality, we can easily derive the following hierarchy of lower bounds:

$$I(X;Y) \geq I_{a\text{-DV}}^{\mathscr{H}}(X;Y) \geq I_{\eta\text{-DV}}^{\mathscr{H}}(X;Y) \geq I_{f\text{-div}}^{\mathscr{H}}(X;Y). \tag{21}$$

In Theorem 2, we apply the $\eta$-trick of Lemma 1 to $I_{a\text{-DV}}^{\mathscr{H}}(X;Y)$, where we replace each $\log\left(\mathbb{E}_{p_x} e^{f(x,y)}\right)$ using the variational form of log which leads to $\eta(y)$. The interchangeability between minimization and expectation and going from scalar $\eta$ for each $y$ to a function $\eta(.)$ needs some care and we prove it in the Appendix B.1, appealing to the interchangeability principle (Lemma 1 in Dai et al. (2016)).

**Theorem 2** ($\eta$-trick for Improved DV Bound). *Let $\mathscr{H}$ be a space of bounded functions defined on $\mathcal{X} \times \mathcal{Y}$ we have: $I(X;Y) \geq -\inf_{f \in \mathscr{H}, \eta \in \mathcal{G}(\mathcal{Y})} \mathbb{E}_{p_x \cdot p_y} e^{f(x,y)-\eta(y)} + \mathbb{E}_{p_y} \eta(y) - 1 - \mathbb{E}_{p_{x,y}} (f(x,y))$, where $\mathscr{G}(\mathcal{Y})$ is the entire space of functions defined on $\mathcal{Y}$. Let $\mathscr{H}_\eta$ be a function space subset of $\mathscr{G}(\mathcal{Y})$ we obtain:*

$$I(X;Y) \geq I_{a\text{-}DV}^{\mathscr{H}}(X;Y) = \sup_{f \in \mathscr{H}, \eta(.) \in \mathscr{H}_\eta} \mathbb{E}_{p_{x,y}} f(x,y) - \mathbb{E}_{p_x \cdot p_y} e^{f(x,y)-\eta(y)} - \mathbb{E}_{p_y} \eta(y) + 1. \quad (22)$$

Hence, we recover the lower bound given in Poole et al. (2019) (given in Table 1), that was derived through other variational principles. In the following, we make a few remarks about $I_{a\text{-}DV}^{\mathscr{H}}(X;Y)$:

1. The $I_{a\text{-}DV}^{\mathscr{H}}(X;Y)$ bound, as given in Poole et al. (2019), can be also obtained by setting $\eta(y) = \log(a(y)), a(y) > 0$. While this seems equivalent, the restriction of $a$ to $\mathbb{R}^+$ ($\eta$ is in $\mathbb{R}$) changes the landscape of the optimization problem and the interchangeability principle does not apply in a straightforward way.

2. Poole et al. (2019) refers to $a(y)$ as a "critic" (as in the reinforcement learning literature). As shown in our analysis in Section 3, when we restrict $\mathscr{H}$ to an RKHS, $\eta(y)$ plays a role of a Lagrangian multiplier or a dual potential in a ratio estimation problem. In this case it is enforcing multiple constraints $\int_{\mathcal{X}} r(x,y)p(x)dx = 1$ for each $y$, instead of imposing a single constraint $\int_{\mathcal{X}} \int_{\mathcal{Y}} r(x,y)p(x)p(y)dxdy = 1$ as in the $I_{\eta\text{-}DV}^{\mathscr{H}}(X;Y)$ bound.

3. No Free Lunch: while $I_{a\text{-}DV}^{\mathscr{H}}(X;Y)$ is tighter than our $I_{\eta\text{-}DV}^{\mathscr{H}}(X;Y)$ bound, it comes at the price of estimating a function $\eta(y)$ together with the witness function $f$. This triggers a higher computational cost and larger estimation errors.

4. We show in Appendix C how $I_{a\text{-}DV}^{\mathscr{H}}(X;Y)$ can be extended to conditional MI estimation.

5. Generality: **$\boldsymbol{\eta}$-DV** bound is applicable for general probability distributions $\mathbb{P}$ and $\mathbb{Q}$: $\text{KL}(\mathbb{P}, \mathbb{Q}) \geq D_{\eta\text{-}DV}^{\mathscr{H}}(\mathbb{P}, \mathbb{Q})$, while $I_{a\text{-}DV}^{\mathscr{H}}(X;Y)$ only works for $\mathbb{P} = p_{xy}$ and $\mathbb{Q} = p_x p_y$.

Finally, in Algorithm 1 we outline the steps of $\eta$-MINE for MI estimation.

---

**Algorithm 1** $\eta$-MINE (Stochastic BCD )

---

**Inputs:** $X, Y$ dataset $X \in \mathbb{R}^{N \times d_x}, Y \in \mathbb{R}^{N \times d_y}$, such that $(x_i = X_{i,.}, y_i = Y_{i,.}) \sim p_{xy}$
**Hyperparameters:** $\alpha_\eta, \alpha_\theta$ (learning rates), $n_c$ (number of critic updates)
**Initialize** $\eta, \theta$ parameter of the neural network $f_\theta$
**for** $i = 1 \ldots$ Maxiter **do**
    **for** $j = 1 \ldots n_c$ **do**
        Fetch a minibatch of size $N$ $(x_i, y_i) \sim p_{xy}$
        Fetch a minibatch of size $N$ $(x_i, \tilde{y}_i) \sim p_x p_y$ {$\tilde{y}_i$ obtained by permuting rows of $Y$}
        Evaluate $\hat{L}(f_\theta, \eta) = e^{-\eta} \frac{1}{N} \sum_{i=1}^N e^{f_\theta(x_i, \tilde{y}_i)} - \frac{1}{N} \sum_{i=1}^N f_\theta(x_i, y_i) + \eta - 1$
        Stochastic Gradient step on $\theta$:
        $\theta \leftarrow \theta - \alpha \frac{\partial \hat{L}(f_\theta, \eta)}{\partial \theta}$ {We use ADAM}
    **end for**
    Update $\eta$:
    $\eta \leftarrow \eta - \alpha_\eta \frac{\partial \hat{L}(f_\theta, \eta)}{\partial \eta}$
**end for**
**Output:** $f_\theta, \eta, \hat{I}_{\eta\text{-}DV}^{\mathscr{H}}(X, Y) = \frac{1}{N} \sum_{i=1}^N f_\theta(x_i, y_i) - e^{-\eta} \frac{1}{N} \sum_{i=1}^N e^{f_\theta(x_i, \tilde{y}_i)} - \eta + 1$

---

## 5 EXPERIMENTS

In this Section, we conduct a set of experiments using synthetic and real data on a number of applications to compare the proposed $\eta$-MINE MI estimator to the existing baselines.

**MI estimation.** We compared different MI estimators on three synthetically generated Gaussian datasets [5K training and 1K testing samples]. Each evaluated MI estimator was run 10 times and its average performance (together with the standard deviation) is shown in Fig. 3.a. As can be seen,

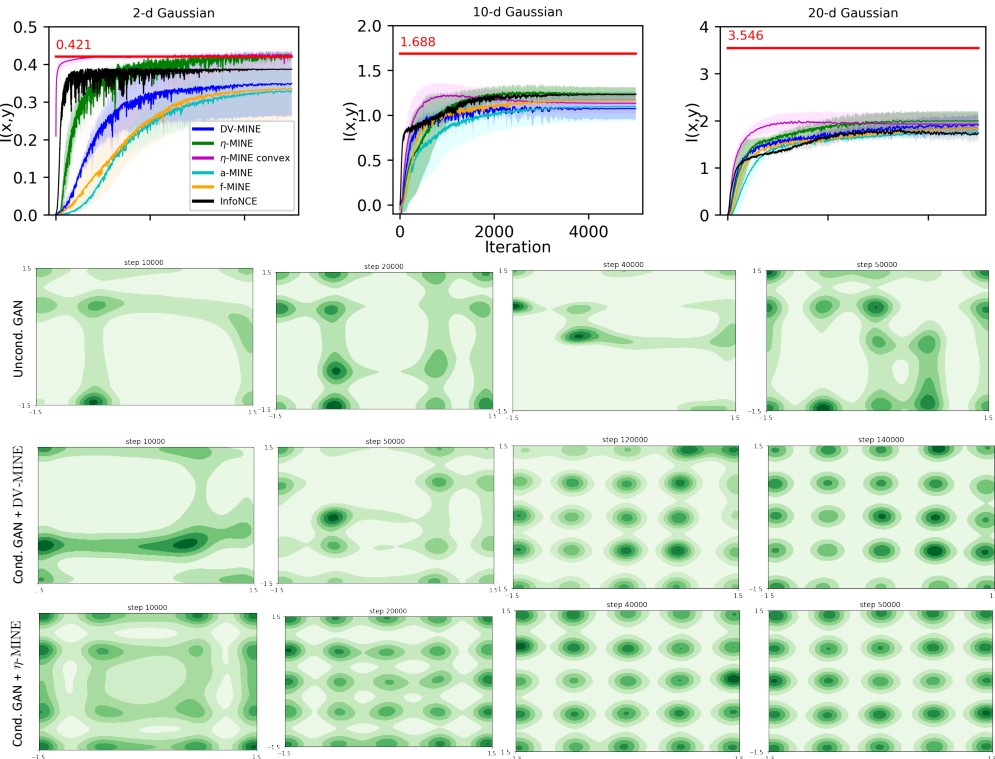

**Figure 3:** Performance of different MI estimators on synthetic Gaussian datasets. Top: *MI estimation*. Data was sampled from 2-, 10- and 20-dimensional Gaussian distributions with randomly generated means and covariance matrices. As we increase the data complexity, the difference between estimators decreases, although we observed that $\eta$-MINE (or its convex extension) on average performed better than the baseline methods, converging to the true MI [red line] faster. Bottom: *MI for GAN regularization*. Top row: unconditional GAN baseline fails at capturing all 25 modes in the Gaussian mixture. Middle row: MI-regularized conditional GAN using DV-MINE (Belghazi et al., 2018) converges after 140K steps of the generator. We found this estimator to be sensitive to the hyper-parameters and unstable during training. Bottom row: MI-regularized conditional GAN using $\eta$-MINE; the model converges in 50K steps of the generator.

MI estimation in high dimensions is a challenging task, where the estimation error for all methods increases as the data dimensionality grows [red line shows true value for MI]. Nevertheless, the proposed $\eta$-MINE is able to achieve on average more accurate results compared to the existing baselines. We also see that the convex formulation of $\eta$-MINE has overall a better performance and fast convergence rate. This estimator has a linear witness function that is defined as $f(\cdot) = \langle w, \Phi(\cdot) \rangle$ using pre-trained fixed feature map $\Phi(\cdot)$ from the regular $\eta$-MINE. In the experiments that follow we compare the proposed estimator $\eta$-MINE to DV-MINE, as a main baseline approach.

**MI-regularized GAN.** We investigate GAN training improvements with MI, and especially diminishing mode collapse, as addressed in Belghazi et al. (2018) in Section 5.1. Belghazi et al. (2018) uses a 25-Gaussian dataset to show improvements on GAN Clustering by using MI objective for regularization. As in InfoGAN (Chen et al., 2016), the conditional generator $G$ is supplied with random codes $c$ along with noise $z$, we maximize the mutual information between $I(G(z, c), c)$ using $\eta$-MINE estimators. In Fig. 3.b, we repeat this task and establish that $\eta$-MINE can recover all the modes within a fewer steps than DV-MINE (50K vs. 140K) and with a stable training.

**Self-supervised: Deep InfoMax** (Hjelm et al., 2018). In unsupervised or self-supervised learning, the objective is to build a model without relying on labeled data but using an auxiliary task to learn informative features that can be useful for various downstream tasks. In this experiment we evaluate the effectiveness of the proposed $\eta$-MINE estimator for unsupervised feature learning using the recently proposed Deep InfoMax method from Hjelm et al. (2018). For feature representation we used an encoder similar to DCGAN (Radford et al., 2015), shown in Fig. 4.a and evaluated results on CIFAR10 and STL10 datasets (STL10 images were scaled down to match CIFAR10 resolution).

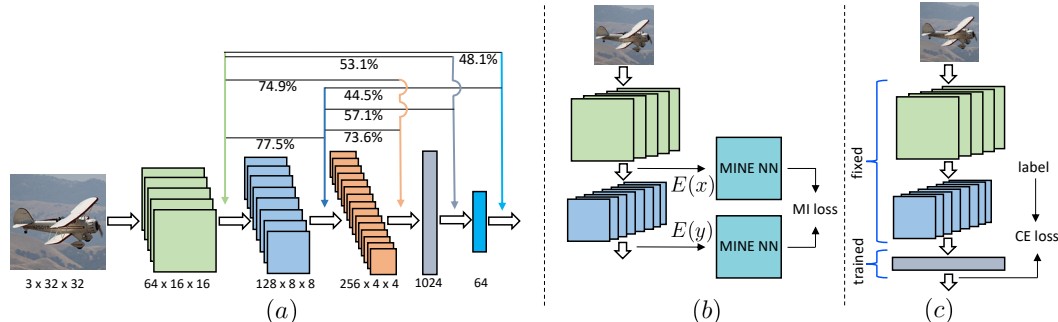

**Figure 4:** (a) Encoder architecture and classification accuracy (top1) on CIFAR-10 dataset for different pre-trained encoders. Each number represents test accuracy of the system trained by maximizing MI between features from layers pointed by the corresponding arrows. Interestingly, the highest accuracy was obtained by pre-training encoder composed of just the first two convolutional layers (see (b) and (c) for details of this process). (b) Model pre-training by maximizing MI between features from different layers [additionally transformed by a neural net, aimed at constructing witness function $f(\cdot)$]. (c) After pre-training, we fix encoder and attach a trainable linear layer to perform traditional supervised training on the same or different datasets.

The encoder is trained by maximizing MI $I(x', y')$ between features from one of the shallow layers $l$ ($x' = E_l(x)$) and a deeper layer $k$ ($y' = E_k(y)$). We examined different layer combinations and found that the encoder composed of only the first two convolutional layers give the best performance on the downstream tasks. As shown in Fig. 4.b the encoder features are passed through additional trainable neural network layers, whose job is to build a classifier $f(x', y')$, discriminating cases when $x'$ are $y'$ are coming from the same image and cases when $x'$ are $y'$ are unrelated. Finally, we attach a linear layer to the pre-trained and now fixed encoder (see Fig. 4.c) to perform supervised training. In Tab. 2 we present the results

**Table 2:** Classification accuracy (top1) results on CI-FAR10 (C10) and STL10 (S10) for unsupervised pre-training task with DV-MINE and $\eta$-MINE using encoder in Fig. 4.b. For reference we also list results for supervised (CE) training using full encoder in Fig. 4.a. $\eta$-MINE-based pre-training achieves overall better results when evaluated on same or different data, even outperforming the supervised model on S10.

| | | Test | | | | |
| | | C10 | | | S10 | |
| Train | DV | $\eta$ | Sup. | DV | $\eta$ | Sup. |
|---|---|---|---|---|---|---|
| C10 | **77.5** | 74.8 | 84.2 | 55.1 | **56.4** | - |
| S10 | 67.5 | **68.3** | - | 61.8 | **63.3** | 61.3 |

for two MI estimators: $\eta$-MINE and DV-MINE, whose loss functions are listed in Tab. 1. As can be seen, $\eta$-MINE-based pre-training performs competitively with DV-MINE, achieving overall better results on both datasets, showing practical benefits of the proposed approach.

**Self-Supervised: Jigsaws with MI**. The self-supervision *Jigsaw* pretext task (Noroozi & Favaro, 2016; Kolesnikov et al., 2019) aims at solving an image jigsaw puzzle by predicting the scrambling permutation $\pi$. From image $X$ with $x = \{x_1 \ldots x_9\}$ 3×3 jigsaw patches, and permutation $y = \pi_i$: $[1, 9] \rightarrow [1, 9]$, scrambled patches $x_{\pi_i} = \{x_{\pi_i(1)} \ldots x_{\pi_i(9)}\}$ generate the puzzle. Each patch is passed through an encoder $E$, which needs to learn meaningful representations such that a classifier $C_J$ can solve the puzzle by predicting the scrambling order $\pi_i$ (Noroozi & Favaro, 2016) (see Fig. 5.a). While this standard formulation relies on a CE-based classification of the permutation, we propose here to use MI Jigsaw, where we train the encoder $E$ to maximize $I(E(x_{\pi_i}); \pi_i)$ by using MI estimators DV- and $\eta$-MINE, as seen in Fig. 5.b. A patch preprocessing similar to Kolesnikov et al. (2019) avoids shortcuts based on color aberration, patch edge matching, etc. (Noroozi & Favaro, 2016); for details of our implementation, see Appendix D. All our models are built on a 10% subset of ImageNet (128K train., 5K val., 1K classes) as proposed by Kolesnikov et al. (2019). This is a larger set than Tiny ImageNet (200 classes) used in many publications. $E$ is a ResNet50 for all our experiments. In our ImageNet target classification task, $E$ from CE and MI Jigsaw trainings are frozen (at 200 epochs) and followed by linear classifier $C$ (Fig. 5.c); an adequate setup for comparing encoders as argued by Kolesnikov et al. (2019).

All models are compared on target task in Tab. 3. Best accuracy numbers are reported for $C$s trained for exactly 200 epochs. For all results, $E$ is trained from Jigsaw task (CE or MI) and frozen, with *only* $C$ trained as in Kolesnikov et al. (2019). DV- and $\eta$-MINE share the same $f$ architecture. $\eta$-MINE gives better accuracy performance compared to DV-MINE. CE model trained from a Jigsaw-

supervised encoder $E$ provides an upper-bound for supervised performance for $E$ from the Jigsaw task. Despite CE being better than $\eta$-MINE and DV-MINE, $\eta$-MINE does a respectable job at learning a useful representation space for the target task, better than DV-MINE. More details about the Jigsaw experiments setup can be found in Appendix D.

**Table 3:** ImageNet (10% subset) classification accuracy results (in %). DV-MINE and $\eta$-MINE are using fixed Encoder from MI training. CE is for using a CE Jigsaw Encoder. Numbers reported are average and standard deviations over 8 models from different initialization seeds.

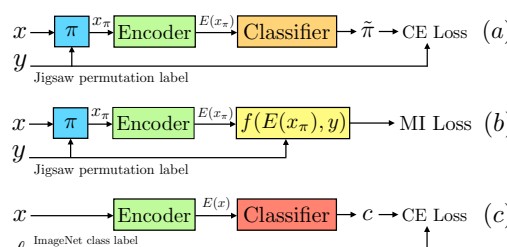

**Figure 5:** (a) Jigsaw CE training. (b) Jigsaw MI training. (c) ImageNet Classification CE training.

|  | DV-MINE | $\eta$-MINE | CE |
|---|---|---|---|
| top1 | $8.5 \pm 1.3$ | $\mathbf{11.0 \pm 1.1}$ | $12.9 \pm 0.3$ |
| top5 | $20.0 \pm 2.5$ | $\mathbf{24.1 \pm 2.2}$ | $28.1 \pm 0.6$ |
| top10 | $27.8 \pm 3.1$ | $\mathbf{32.7 \pm 2.2}$ | $37.7 \pm 0.6$ |

## 6 CONCLUSION

In this paper we introduced a new lower bound on the KL divergence and showed how it can be used to improve the estimation of mutual information using neural networks. Theoretically we proved that the dual of our $\eta$-DV formulation reduces to a constrained likelihood ratio estimation. In practice, the stability of $\eta$-MINE is due to its unbiased gradient. We tested our estimator $\eta$-MINE on synthetic data and applied it on various real-world tasks where MI can be used as a regularizer, or as an objective in self-supervised learning. We used $\eta$-MINE in unsupervised learning of representations through MI maximization and by solving Jigsaw puzzles. We leave for a future work the analysis and various possible applications of conditional mutual information neural estimators.

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

**Supplementary Material for Improved Mutual Information Estimation**

The supplementary material is organized as follows:

- In Appendix A we layout proofs of Lemmas 1, 2 and 3. Those lemmas provide the $\eta$-DV formulation. We also give in this Appendix the proof of the main Theorem 1. This theorem gives the dual formulation of this variational bound in RKHS with Tikhonov Regularization and sheds the light on the ratio estimation interpretation of $\eta$-DV.

- We give in Appendix A.1 the impact of other regularizers on the estimation of KL divergence in RKHS.

- In Appendix B, we show that further improved bounds can be obtained for the particular case of mutual information leading to an improved Donsker-Varadhan. We show that by using the interchangeability principle, we can obtain an equivalent functional $\eta$-DV bound that is related to $a$-MINE introduced in Poole et al. (2016).

- In Appendix C, we show how those improved variational bounds can be extended to conditional mutual information.

- In Appendix D, we give more details on the experimental setup of self-supervised learning with the jigsaw task.

## A    TECHNICAL PROOFS

*Proof of Lemma:1.* Let $g(\eta) = e^{-\eta}x + \eta - 1$. We have $g'(\eta) = -e^{-\eta}x + 1$ and $g''(\eta) = e^{-\eta}x > 0$. Hence $g$ is convex and admits a unique minimum. First order optimality gives us: $g'(\eta^*) = -e^{-\eta^*}x + 1 = 0$, hence $e^{-\eta^*}x = 1$ and $\eta^* = \log(x)$. Finally, we get

$$g(\eta^*) \quad = \quad e^{-\log(x)}x + \log(x) - 1 = e^{\log(\frac{1}{x})}x + \log(x) - 1 = 1 + \log(x) - 1 = \log(x).$$

$\square$

*Proof of Lemma 2.* By Donsker-Varadhan representation we have:

$$\begin{aligned}
D_{\mathrm{DV}}^{\mathscr{H}}(\mathbb{P}, \mathbb{Q}) &= \sup_{f \in \mathscr{H}} \mathbb{E}_{x \sim \mathbb{P}} f(x) - \log\left(\mathbb{E}_{x \sim \mathbb{Q}} e^{f(x)}\right) \\
&= -\inf_{f \in \mathscr{H}} \log\left(\mathbb{E}_{x \sim \mathbb{Q}} e^{f(x)}\right) - \mathbb{E}_{x \sim \mathbb{P}} f(x) \\
&= -\inf_{f \in \mathscr{H}} \{\inf_{\eta} e^{-\eta} \mathbb{E}_{x \sim \mathbb{Q}} e^{f(x)} + \eta - 1\} - \mathbb{E}_{x \sim \mathbb{P}} f(x) \\
&= -\inf_{f \in \mathscr{H}, \eta \in \mathbb{R}} \{e^{-\eta} \mathbb{E}_{x \sim \mathbb{Q}} e^{f(x)} - \mathbb{E}_{x \sim \mathbb{P}} f(x) + \eta - 1\}.
\end{aligned}$$

$\square$

*Proof of Lemma 3.* $\mathscr{L}$ is a sum of a linear function in $w$, $\eta$ and $\Omega(w)$, that are both convex. We need to prove that $f_b(w, \eta) = e^{-\eta} e^{\langle w, b \rangle}$ is jointly convex, since $\mathscr{L}$ can be written as a sum of $f_b$ and other convex functions:

$$\nabla_w f_b(w, \eta) = e^{-\eta} e^{\langle w, b \rangle} b, \quad \nabla_w^2 f_b(w, \eta) = e^{-\eta} e^{\langle w, b \rangle} b \otimes b \text{ PSD (Positive Semi-Definite)}$$

$$\frac{\partial^2 f_b}{\partial w \partial \eta} = -e^{-\eta} e^{\langle w, b \rangle} b, \quad \frac{\partial^2 f_b}{\partial \eta^2} = e^{-\eta} e^{\langle w, b \rangle}.$$

The Hessian of $f_b(w, \eta)$, $H f_b$, has the following form:

$$H f_b(w, \eta) = \begin{bmatrix} \frac{\partial^2 L}{\partial w \otimes \partial w} & \frac{\partial^2 L}{\partial w \partial \eta} \\ \frac{\partial^2 L}{\partial \eta \partial w} & \frac{\partial^2 L}{\partial \eta^2} \end{bmatrix} = \begin{bmatrix} e^{-\eta} e^{\langle w, b \rangle} b \otimes b & -e^{-\eta} e^{\langle w, b \rangle} b \\ -e^{-\eta} e^{\langle w, b \rangle} b^{\top} & e^{-\eta} e^{\langle w, b \rangle} \end{bmatrix}$$

Let us prove that for all $(w, \eta)$: $(w', \eta')^{\top} H f_b(w, \eta)(w', \eta') \geq 0, \forall (w', \eta')$. We have:

$$\begin{aligned}
(w', \eta')^{\top} H f_b(w, \eta)(w', \eta') &= e^{-\eta} e^{\langle w, b \rangle} (\langle w', b \rangle)^2 - 2\eta' e^{-\eta} e^{\langle w, b \rangle} \langle w', b \rangle + \eta'^2 e^{-\eta} e^{\langle w, b \rangle} \\
&= e^{-\eta} e^{\langle w, b \rangle} \left((\langle w', b \rangle)^2 - 2\eta' \langle w', b \rangle + \eta'^2\right) \\
&= e^{-\eta} e^{\langle w, b \rangle} (\langle w', b \rangle - \eta')^2 \geq 0.
\end{aligned}$$

Hence, $f_b$ is jointly convex in $\eta$ and $w$, and so is $L$. Now adding the perturbation $\varepsilon e^{-\eta}$ to the loss, we get

$$w', \eta')^\top H f_b(w, \eta)(w', \eta') = e^{-\eta} e^{\langle w, b \rangle} \left( \langle w', b \rangle - \eta' \right)^2 + \varepsilon e^{-\eta} > 0,$$

for finite $\eta$ and hence we have a strict convexity. $\qquad \square$

*Proof of Theorem 1.* Let's consider:

$$f(x) = \langle w, \Phi(x) \rangle, \text{ and } \varphi(t) = e^t.$$

Recall that the Fenchel conjugate of $f$ is $f^*(p) = \sup_x \langle p, x \rangle - f(x)$. It follows that $\varphi^*(t) = t \log(t) - t$ if $t > 0$, 0 for $t = 0$, and $\infty$ otherwise.

$$\min_{w, \eta} e^{-\eta} \mathbb{E}_{y \sim \mathbb{Q}} \varphi \left( \langle w, \Phi(y) \rangle \right) - \mathbb{E}_{x \sim \mathbb{P}} \langle w, \Phi(x) \rangle + \eta - 1 + \Omega(w)$$

$$= \min_{w, \eta} e^{-\eta} \int_{\mathcal{X}} \varphi \left( \langle w, \Phi(y) \rangle \right) q(y) dy - \left\langle w, \int_{\mathcal{X}} \Phi(x) p(x) dx \right\rangle + \eta - 1 + \Omega(w)$$

$$= \min_{w, \eta, z} e^{-\eta} \int_{\mathcal{X}} \varphi \left( z(y) \right) q(y) dy - \left\langle w, \int_{\mathcal{X}} \Phi(x) p(x) dx \right\rangle + \eta - 1 + \Omega(w) \quad \text{s.t. } z(y) = \langle w, \Phi(y) \rangle$$

Introducing Lagrangian's $\alpha$ we have

$$= \min_{w, \eta, z} \max_{\alpha} e^{-\eta} \int_{\mathcal{X}} \varphi \left( z(y) \right) q(y) dy - \left\langle w, \int_{\mathcal{X}} \Phi(x) p(x) dx \right\rangle + \eta - 1 + \Omega(w) - \int_{\mathcal{X}} \alpha(y)(\langle w, \Phi(y) \rangle - z(y)) dy$$

$$= \max_{\alpha} \min_{w, \eta, z} e^{-\eta} \int_{\mathcal{X}} \varphi \left( z(y) \right) q(y) dy - \left\langle w, \int_{\mathcal{X}} \Phi(x) p(x) dx \right\rangle + \eta - 1 + \Omega(w) - \int_{\mathcal{X}} \alpha(y)(\langle w, \Phi(y) \rangle - z(y)) dy$$

using strong duality (convex function and lower semi-continuous) to swap min, max (Ekeland & Turnbull, 1983)

$$= \max_{\alpha} \min_{\eta} - \left( \max_{z} - \left( \int_{\mathcal{X}} (\varphi \left( z(y) \right) q(y) e^{-\eta} + \alpha(y) z(y)) dy \right) \right)$$

$$- \left( \max_{w} \left\langle w, \int_{\mathcal{X}} \Phi(x) p(x) dx + \int_{\mathcal{X}} \alpha(y) \Phi(y) dy \right\rangle - \Omega(w) - \eta + 1 \right)$$

$$= \max_{\alpha} \min_{\eta} - \left( \int_{\mathcal{X}} q(y) e^{-\eta} \max_{z} -\varphi(z) - \frac{\alpha(y)}{q(y) e^{-\eta}} z \right) - \Omega^* \left( \int_{\mathcal{X}} \Phi(x) p(x) dx + \int_{\mathcal{X}} \alpha(y) \Phi(y) dy \right) + \eta - 1$$

$$= \max_{\alpha < 0} \min_{\eta} - \left( \int_{\mathcal{X}} q(y) e^{-\eta} \varphi^*(-\frac{\alpha(y)}{q(y) e^{-\eta}}) \right) - \Omega^* \left( \int_{\mathcal{X}} \Phi(x) p(x) dx + \int_{\mathcal{X}} \alpha(y) \Phi(y) dy \right) + \eta - 1$$

$$= \max_{\alpha > 0} \min_{\eta} - \left( \int_{\mathcal{X}} q(y) e^{-\eta} \varphi^*(\frac{\alpha(y)}{q(y) e^{-\eta}}) \right) - \Omega^* \left( \int_{\mathcal{X}} \Phi(x) p(x) dx - \int_{\mathcal{X}} \alpha(y) \Phi(y) dy \right) + \eta - 1$$

$$= - \min_{\alpha > 0} \max_{\eta} \left( \int_{\mathcal{X}} q(y) e^{-\eta} \varphi^*(\frac{\alpha(y)}{q(y) e^{-\eta}}) \right) + \Omega^* \left( \int_{\mathcal{X}} \Phi(x) p(x) dx - \int_{\mathcal{X}} \alpha(y) \Phi(y) dy \right) - \eta + 1$$

Hence, we have for $\alpha > 0$, and finite $\eta$:

$$\int_{\mathcal{X}} q(y) e^{-\eta} \varphi^* \left( \frac{\alpha(y)}{q(y) e^{-\eta}} \right) dy = \int_{\mathcal{X}} q(y) e^{-\eta} \left( \frac{\alpha(y)}{q(y) e^{-\eta}} \log(\frac{\alpha(y)}{q(y) e^{-\eta}}) - \frac{\alpha(y)}{q(y) e^{-\eta}} \right) dy$$

$$= \int_{\mathcal{X}} q(y) e^{-\eta} \left( \frac{\alpha(y)}{q(y) e^{-\eta}} \log(\frac{\alpha(y)}{q(y) e^{-\eta}}) - \frac{\alpha(y)}{q(y) e^{-\eta}} \right) dy$$

$$= \int_{\mathcal{X}} [\alpha(y) \log \left( \frac{\alpha(y)}{q(y) e^{-\eta}} \right) - \alpha(y)] dy$$

$$= \int_{\mathcal{X}} \alpha(y) \log \left( \frac{\alpha(y)}{q(y)} \right) dy + (\eta - 1) \int \alpha(y)) dy.$$

Define

$$r(y) = \frac{\alpha(y)}{q(y)} > 0$$

then we have

$$(\text{A}) : \int_{\mathcal{X}} q(y)e^{-\eta}\varphi^*\left(\frac{\alpha(y)}{q(y)e^{-\eta}}\right)dy = \int_{\mathcal{X}} r(y)\log\left(r(y)\right)q(y)dy + (\eta-1)\int_{\mathcal{X}} r(y)q(y)dy,$$

and

$$(\text{B}) : \Omega^*\left(\int_{\mathcal{X}} \Phi(x)p(x)dx - \int_{\mathcal{X}} \alpha(y)\Phi(y)dy\right) = \Omega^*\left(\int_{\mathcal{X}} \Phi(x)p(x)dx - \int_{\mathcal{X}} r(y)q(y)\Phi(y)dy\right).$$

Hence, by replacing (A) and (B) by their expressions, we get:

$$\min_{w,\eta}\mathscr{L}(w,\eta)$$

$$= -\min_{\alpha>0}\max_{\eta}\left(\int_{\mathcal{X}} q(y)e^{-\eta}\varphi^*(\frac{\alpha(y)}{q(y)e^{-\eta}})\right) + \Omega^*\left(\int_{\mathcal{X}} \Phi(x)p(x)dx - \int_{\mathcal{X}} \alpha(y)\Phi(y)dy\right) - \eta + 1$$

$$= -\min_{r(.)>0}\max_{\eta}\int_{\mathcal{X}} r(y)\log\left(r(y)\right)q(y)dy + (\eta-1)\left(\int_{\mathcal{X}} r(y)q(y)dy - 1\right)$$

$$+ \Omega^*\left(\int_{\mathcal{X}} \Phi(x)p(x)dx - \int_{\mathcal{X}} r(y)q(y)\Phi(y)dy\right)$$

Now consider again the $\eta$-DV Bound:

$$-\min_{w,\eta}\mathscr{L}(w,\eta)$$

$$= \min_{r>0}\max_{\eta}\int_{\mathcal{X}} r(y)\log\left(r(y)\right)q(y)dy + (\eta-1)(\int_{\mathcal{X}} r(y)q(y)dy - 1)$$

$$+ \Omega^*\left(\int_{\mathcal{X}} \Phi(x)p(x)dx - \int_{\mathcal{X}} r(y)q(y)\Phi(y)dy\right)$$

$$= \min_{r>0}\int_{\mathcal{X}} r(y)\log\left(r(y)\right)q(y)dy + \Omega^*\left(\int_{\mathcal{X}} \Phi(x)p(x)dx - \int_{\mathcal{X}} r(y)q(y)\Phi(y)dy\right)$$

$$\text{s.t. } \int_{\mathcal{X}} r(y)q(y)dy = 1.$$

We see that $\eta$-DV amounts to a likelihood ratio $r$ estimation, where we match with the dual norm $\Omega^*$ the difference of kernel mean embedding of $p$, and $rq$. And $\eta$ plays a role of a Lagrange multiplier that enforces the constraint that $rq$ is a normalized distribution. In particular, consider $\Omega(u) = \frac{\lambda}{2}||u||_2^2$, so that $\Omega^*(v) = \frac{1}{2\lambda}||v||^2$, and hence we have the following equivalent problem of the regularized $\eta$-DV bound:

$$\min_{r>0}\int_{\mathcal{X}} r(y)\log\left(r(y)\right)q(y)dy + \frac{1}{2\lambda}\underbrace{\left\|\int_{\mathcal{X}} \Phi(x)p(x)dx - \int_{\mathcal{X}} r(y)q(y)\Phi(y)dy\right\|_2^2}_{\text{MMD matching of } rq \text{ and } p}$$

$$\text{s,t. } \underbrace{\int_{\mathcal{X}} r(y)q(y)dy = 1.}_{\text{Normalization Constraint}}$$

Another way of coming informally to the regularized $\eta$-DV objective is to start from expression of KL using $r$:

$$\min_{r>0}\int_{\mathcal{X}} r(y)\log(r(y))q(y)dx$$

$$\text{MMD}_\Phi(rq,p) < \varepsilon$$

$$\int_{\mathcal{X}} r(x)q(x) = 1.$$

We see that $r$ in the dual function in estimating the ratio, $\eta$ is the Lagrangian of the inequality constraint, and the MMD is added to the cost as a penality. $\qquad\square$

A.1 IMPACT OF REGULARIZERS ON KL ESTIMATION

As discussed before, the choice of the regularizer $\Omega$ translates to a particular type of metric used in estimating the likelihood ratio: $\Omega(.) = \|.\|_2^2$ translates in a ratio estimation in the MMD sense. Here, we consider other choices of regularization used in Fisher discriminant analysis (Harchaoui et al., 2008; Mroueh & Sercu, 2017), specifically:

$$\boldsymbol{\eta}\text{-}\mathbf{DV}\text{–Fisher} : \min_{f,\eta} L(f,\eta) + \frac{\lambda}{2}\mathbb{E}_{x\sim\frac{\mathbb{P}+\mathbb{Q}}{2}} f^2(x), \qquad (23)$$

or gradient penalty regularizer (Gulrajani et al., 2017; Mroueh et al., 2018; 2019):

$$\boldsymbol{\eta}\text{-}\mathbf{DV}\text{–Sobolev} : \min_{f,\eta} L(f,\eta) + \frac{\lambda}{2}\mathbb{E}_{x\sim\frac{\mathbb{P}+\mathbb{Q}}{2}} \|\nabla_x f(x)\|^2. \qquad (24)$$

It is easy to see that when we restrict $f$ to $\mathscr{H}$, $\boldsymbol{\eta}$-$\mathbf{DV}$–Fisher has the same objective as in Eq. 11 with $\Omega(w) = \frac{\lambda}{2}\langle w, (C(\mathbb{P},\mathbb{Q}))w\rangle$, where $C(\mathbb{P},\mathbb{Q}) = \mathbb{E}_{\frac{\mathbb{P}+\mathbb{Q}}{2}}\Phi(x)\otimes\Phi(x)$ is a covariance matrix. Similarly, $\boldsymbol{\eta}$-$\mathbf{DV}$–Sobolev corresponds to Eq. 11 with $\Omega(w) = \frac{\lambda}{2}\langle w, (D(\mathbb{P},\mathbb{Q}))w\rangle$, where $D(\mathbb{P},\mathbb{Q}) = \mathbb{E}_{(\mathbb{P}+\mathbb{Q})/2}\sum_{j=1}^d \partial_j\Phi(x)\otimes\partial_j\Phi(x)$ is a Gramian of derivatives. The convex conjugate $\Omega^*(w)$ is therefore $\frac{1}{2\lambda}\langle w, (C^\dagger(\mathbb{P},\mathbb{Q}))w\rangle$ and $\frac{1}{2\lambda}\langle w, (D^\dagger(\mathbb{P},\mathbb{Q}))w\rangle$, for $\boldsymbol{\eta}$-$\mathbf{DV}$–Fisher and $\boldsymbol{\eta}$-$\mathbf{DV}$–Sobolev respectively (where $\dagger$ denotes the pseudo-inverse).

Let $\delta_{\mathbb{P},\mathbb{Q}} = \mathbb{E}_{\mathbb{P}}\Phi(x) - \mathbb{E}_{\mathbb{Q}}\Phi(y)$. Recall from Harchaoui et al. (2008) that the Fisher discrepancy is $\mathscr{F}_\Phi^2(\mathbb{P},\mathbb{Q}) = \langle\delta_{\mathbb{P},\mathbb{Q}}, C(\mathbb{P},\mathbb{Q})^\dagger\delta_{\mathbb{P},\mathbb{Q}}\rangle$, and from Mroueh et al. (2018; 2019) the Sobolev discrepancy is $\mathcal{S}_\Phi^2(\mathbb{P},\mathbb{Q}) = \langle\delta_{\mathbb{P},\mathbb{Q}}, D(\mathbb{P},\mathbb{Q})^\dagger\delta_{\mathbb{P},\mathbb{Q}}\rangle$. Now replacing $\Omega^*$ by its expression in Theorem 1 we see that the choice of the regularizer dictates the discrepancy used for the ratio estimation:

$$\boldsymbol{\eta}\text{-}\mathbf{DV}\text{-Fisher} : \min_{r>0}\int_{\mathcal{X}} r(y)\log(r(y))\,q(y)dy + \frac{1}{2\lambda}\mathscr{F}_\Phi^2(rq, p),\ \text{subject to:}\ \int_{\mathcal{X}} r(y)q(y)dy = 1,$$

$$\boldsymbol{\eta}\text{-}\mathbf{DV}\text{-Sobolev} : \min_{r>0}\int_{\mathcal{X}} r(y)\log(r(y))\,q(y)dy + \frac{1}{2\lambda}\mathcal{S}_\Phi^2(rq, p),\ \text{subject to:}\ \int_{\mathcal{X}} r(y)q(y)dy = 1.$$

# B FURTHER BOUNDING MUTUAL INFORMATION: $a$-MINE FORMULATION FROM INTERCHANGEABILITY PRINCIPLE

We start by the classical result of Donsker and Varadhan:

**Lemma 5** (Donsker-Varadhan). *For any Distribution $\mathbb{P}$ and any distribution $\mathbb{Q}$ and any non-negative function $g$ for which $\mathbb{E}_{x\sim\mathbb{P}}\log g(x)$ is finite, we have:*

$$KL(\mathbb{P},\mathbb{Q}) + \log(\mathbb{E}_{x\sim\mathbb{Q}}g(x)) \geq \mathbb{E}_{x\sim\mathbb{P}}\log(g(x)).$$

*This gives us a representation of KL divergence:*

$$KL(\mathbb{P},\mathbb{Q}) = \sup_{g>0}\mathbb{E}_{x\sim\mathbb{P}}\log(g(x)) - \log(\mathbb{E}_{x\sim\mathbb{Q}}g(x)) = \sup_f \mathbb{E}_{x\sim\mathbb{P}}f(x) - \log(\mathbb{E}_{x\sim\mathbb{Q}}e^{f(x)}).$$

The following Lemma shows that a finer application of the Donsker-Varadhan Lemma gives an improved variational representation of MI :

**Lemma 6.** *Let $g > 0$*

$$KL(p_{x,y}, p_x p_y) + \mathbb{E}_{p_x}\log(\mathbb{E}_{p_y}g(x,y)) \geq \mathbb{E}_{p_{x,y}}\log(g(x,y)).$$

*Hence, we get*

$$I(X;Y) = \sup_{g>0}\mathbb{E}_{p_{x,y}}\log(g(x,y)) - \mathbb{E}_{p_x}\log(\mathbb{E}_{p_y}g(x,y))$$

$$= \sup_{f\in\mathscr{G}(\mathcal{X}\times\mathcal{Y})}\mathbb{E}_{p_{x,y}}f(x,y) - \mathbb{E}_{p_x}\log(\mathbb{E}_{p_y}e^{f(x,y)})$$

$$= \sup_{f\in\mathscr{G}(\mathcal{X}\times\mathcal{Y})}\mathbb{E}_{p_{x,y}}f(x,y) - \mathbb{E}_{p_y}\log(\mathbb{E}_{p_x}e^{f(x,y)})$$

*where $\mathscr{G}(\mathcal{X}\times\mathcal{Y})$ is the entire space of functions defined on $\mathcal{X}\times\mathcal{Y}$.*

*Proof.*

$$\text{KL}(p_{x,y}, p_x p_y) + \mathbb{E}_{p_x} \log(\mathbb{E}_{p_y} g(x,y)) = \mathbb{E}_{p_{xy}} \log(\frac{p_{xy}}{p_x p_y}) + \mathbb{E}_{p_x} \log(\mathbb{E}_{p_y} g(x,y))$$

$$= \mathbb{E}_{p_x} \left( \mathbb{E}_{p_{y|x}} \log(\frac{p_{y|x}}{p_y}) + \log(\mathbb{E}_{p_y} g(x,y)) \right)$$

$$= \mathbb{E}_{p_x} \left( \text{KL}(p_{y|x}, p_y) + \log(\mathbb{E}_{p_y} g(x,y)) \right).$$

Now applying Donsker-Varadhan Lemma, we have:

$$\text{KL}(p_{y|x}, p_y) + \log(\mathbb{E}_{p_y} g(x,y)) \geq \mathbb{E}_{p_{y|x}} \log(g(x,y)).$$

We conclude therefore that

$$\text{KL}(p_{x,y}, p_x p_y) + \mathbb{E}_{p_x} \log(\mathbb{E}_{p_y} g(x,y)) \geq \mathbb{E}_{p_x} \left( \mathbb{E}_{p_{y|x}} \log(g(x,y)) \right)$$
$$= \mathbb{E}_{p_{x,y}} \log(g(x,y)).$$

$\square$

## B.1 $\eta$-TRICK AND INTERCHANGEABILITY PRINCIPLE FOR IMPROVED DV

We just showed that an improved Donsker-Varadhan representation of Mutual Information can be be derived as follows:

$$I(X;Y) = \sup_{f \in \mathcal{G}(\mathcal{X} \times \mathcal{Y})} \mathbb{E}_{p_{x,y}} f(x,y) - \mathbb{E}_{p_y} \log(\mathbb{E}_{p_x} e^{f(x,y)}).$$

We show in this section that, by using interchangeability principle, we can replace the "log" non linearity preceding the expectation with an "functional" $\eta$-trick.

We start first by stating the interchangeability principle between expectation and minimum operations:

**Lemma 7** (Interchangeability Principle see Lemma 1 in (Dai et al., 2016)). *Let $\xi$ be a random variable on $\Xi$ and assume for any $\xi \in \Xi$, the function $g(., \xi) : \mathbb{R} \to (-\infty, +\infty)$ is a proper and lower semi-continuous convex function. Then*

$$\mathbb{E}_{\xi} \min_{u \in \mathbb{R}} g(u, \xi) = \min_{u(.) \in \mathcal{G}(\Xi)} \mathbb{E}_{\xi} g(u(\xi), \xi).$$

Applying the interchangeability principle and the variational representation of $\log$ we obtain the following Theorem:

**Theorem 3** (Theorem 2 restated: $\eta$-trick for improved DV Bound). *Let $\mathcal{H}$ be a space of bounded functions defined on $\mathcal{X} \times \mathcal{Y}$, we have:*

$$I(X;Y) \geq - \inf_{f \in \mathcal{H}, \eta \in \mathcal{G}(\mathcal{Y})} \mathbb{E}_{p_x \cdot p_y} e^{f(x,y) - \eta(y)} + \mathbb{E}_{p_y} \eta(y) - 1 - \mathbb{E}_{p_{x,y}} (f(x,y)),$$

*where $\mathcal{G}(\mathcal{Y})$ is the entire space of functions defined on $\mathcal{Y}$. Let $\mathcal{H}_\eta$ be a function space subset of $\mathcal{G}(\mathcal{Y})$, we obtain:*

$$I(X;Y) \geq I_{a\text{-}DV}^{\mathcal{H}}(X;Y) = \sup_{f \in \mathcal{H}, \eta(.) \in \mathcal{H}_\eta} \mathbb{E}_{p_{x,y}} f(x,y) - \mathbb{E}_{p_x \cdot p_y} e^{f(x,y) - \eta(y)} - \mathbb{E}_{p_y} \eta(y) + 1.$$

*Proof.* We know from Lemma 6 that for any function space $\mathcal{H}$

$$I(X;Y) \geq - \inf_{f \in \mathcal{H}} \mathbb{E}_{p_y} \log(\mathbb{E}_{p_x} e^{f(x,y)}) - \mathbb{E}_{p_{x,y}} (f(x,y)). \tag{25}$$

Applying the $\eta$-trick for each $\log$ in the expectation:

$$\mathbb{E}_{p_y} \log(\mathbb{E}_{p_x} e^{f(x,y)}) = \mathbb{E}_{p_y} \min_{\eta \in \mathbb{R}} e^{-\eta} \mathbb{E}_{p_x} e^{f(x,y)} + \eta - 1. \tag{26}$$

We assume here that the function space $f$ is bounded. Before applying the interchangeability Lemma 7, let us verify that all assumptions hold for

$$g(\eta, y) = e^{-\eta}\mathbb{E}_{p_x}e^{f(x,y)} + \eta - 1.$$

**Convex.** $g(., y)$ is convex. Considering derivatives wrt to $\eta$: $g'(\eta, y) = -e^{-\eta}\mathbb{E}_{p_x}(e^{f(x,y)}) + 1$, and $g''(\eta, y) = e^{-\eta}\mathbb{E}_{p_x}(e^{f(x,y)}) > 0$.

**Proper Function.** $g(\eta, .)$ is a proper function, since $f$ is bounded ($-M \le f(x, y) \le M$):

$$g(0, y) = \mathbb{E}_{p_x}e^{f(x,y)} - 1 \le e^M - 1 < \infty.$$

and

$$g(\eta, y) > -\infty, \forall \eta,$$

since

$$g(\eta, y) \ge g(\eta^*, y) = \log(\mathbb{E}_{p_x}e^{f(x,y)}) \ge -M, \forall \eta.$$

**Lower semi-continuity.** $g(\eta, .)$ is lower semi-continuous since it is continuous (sum of continuous functions).

Now back to Eq. 26, we are ready to apply the interchangeability principle given in Lemma 7:

$$
\begin{aligned}
\mathbb{E}_{p_y}\log(\mathbb{E}_{p_x}e^{f(x,y)}) &= \mathbb{E}_{p_y}\min_{\eta\in\mathbb{R}}e^{-\eta}\mathbb{E}_{p_x}e^{f(x,y)} + \eta - 1 \\
&= \mathbb{E}_{p_y}\min_{\eta\in\mathbb{R}}g(\eta, y) \\
&= \min_{\eta(.)\in\mathscr{G}(\mathcal{Y})}\mathbb{E}_{p_y}g(\eta(y), y) \\
&= \min_{\eta(.)\in\mathscr{G}(\mathcal{Y})}\mathbb{E}_{p_y}\left(e^{-\eta(y)}\mathbb{E}_{p_x}e^{f(x,y)} + \eta(y) - 1\right).
\end{aligned}
$$

Using this equivalent form, we have:

$$
\begin{aligned}
&\inf_{f\in\mathscr{H}}\mathbb{E}_{p_y}\log(\mathbb{E}_{p_x}e^{f(x,y)}) - \mathbb{E}_{p_{x,y}}f(x, y) \\
&= \inf_{f\in\mathscr{H}}\min_{\eta(.)\in\mathscr{G}(\mathcal{Y})}\mathbb{E}_{p_y}\left(e^{-\eta(y)}\mathbb{E}_{p_x}e^{f(x,y)} + \eta(y) - 1\right) - \mathbb{E}_{p_{x,y}}(f(x, y) \\
&= \inf_{f\in\mathscr{H}, \eta\in\mathcal{G}(\mathcal{Y})}\mathbb{E}_{p_x.p_y}e^{f(x,y)-\eta(y)} + \mathbb{E}_{p_y}\eta(y) - 1 - \mathbb{E}_{p_{x,y}}(f(x, y).
\end{aligned}
$$

It follows that

$$
\begin{aligned}
I(X; Y) &\ge -\inf_{f\in\mathscr{H}, \eta\in\mathcal{G}(\mathcal{Y})}\mathbb{E}_{p_x.p_y}e^{f(x,y)-\eta(y)} + \mathbb{E}_{p_y}\eta(y) - 1 - \mathbb{E}_{p_{x,y}}(f(x, y) \\
&= \sup_{f\in\mathscr{H}, \eta\in\mathcal{G}(\mathcal{Y})}\mathbb{E}_{p_{x,y}}f(x, y) - \mathbb{E}_{p_x.p_y}e^{f(x,y)-\eta(y)} - \mathbb{E}_{p_y}\eta(y) + 1.
\end{aligned}
$$

Consider now a function space $\mathscr{H}_\eta$ from $\mathcal{Y} \to \mathbb{R}$ we have:

$$I(X; Y) \ge \sup_{f\in\mathscr{H}, \eta\in\mathscr{H}_\eta}\mathbb{E}_{p_{x,y}}f(x, y) - \mathbb{E}_{p_x.p_y}e^{f(x,y)-\eta(y)} - \mathbb{E}_{p_y}\eta(y) + 1,$$

The MI lower bound is looser than regular $\eta-$MINE, since we have two lower bounds passing from all function spaces $\mathscr{G}(\mathcal{X} \times \mathcal{Y})$ on to $\mathscr{H}$, then going from $\mathscr{G}(\mathcal{Y})$ to $\mathscr{H}_\eta$. □

No free lunch: tighter bound but we need to estimate a function $\eta(x)$ which makes bigger estimation errors. To get a good estimate of $\eta(x)$, to obtain its optimal value $\log(\mathbb{E}_{p_y}e^{f(x,y)})$, we need for each $x$ many samples from $p_y$ (which means many epochs of cycling through the same data in random order). The $\eta$-trick with only a scalar variable is better computationally, since it enjoys convexity and has better optimization properties. But as it is clear from this discussion, $\eta$ function of a modality is good for conditional MI estimation which we will discuss next.

## C  CONDITIONAL MUTUAL INFORMATION

To illustrate the power of improved Donsker-Varadhan and its variational representation with the $\eta$-trick and interchangeability principle, we show how to extend this to Conditional Mutual Information estimation.

We have a $(X, Y)$ pair of random variables that may depend on another variable $Z$. This conditional dependence is of high importance in all applications of joint embedding or in the information bottleneck principle, etc.

$$I(X; Y|Z) = \text{KL}(p_{xyz}, p_{x|z}p_{y|z}p(z)),$$

meaning that conditioned on $Z$, how much $X$ and $Y$ are dependent.

Therefore we have:

$$I(X; Y|Z) = \mathbb{E}_{z \sim p_z} \text{KL}(p_{x,y|z}||p_{x|z}p_{y|z}).$$

Now applying Donsker-Varadhan Lemma, we have for all $z$ and for any function $f$

$$\text{KL}(p_{x,y|z}||p_{x|z}p_{y|z}) + \log \mathbb{E}_{p_{x|z}p_{y|z}} f(x, y, z) \geq \mathbb{E}_{p_{x,y|z}} \log(f(x, y, z)).$$

Now marginalizing on $Z$, we get

$$\mathbb{E}_{z \sim p_z} \text{KL}(p_{x,y|z}||p_{x|z}p_{y|z}) + \mathbb{E}_{z \sim p_z} \log \mathbb{E}_{p_{x|z}p_{y|z}} f(x, y, z) \geq \mathbb{E}_{z \sim p_z} \mathbb{E}_{p_{x,y|z}} \log(f(x, y, z)).$$

Hence,

$$I(X; Y|Z) \geq \mathbb{E}_{z \sim p_z} \mathbb{E}_{p_{x,y|z}} \log(f(x, y, z)) - \mathbb{E}_{z \sim p_z} \log \mathbb{E}_{p_{x|z}p_{y|z}} f(x, y, z),$$

and

$$I(X; Y|Z) = \sup_{f \in \mathscr{F}, f > 0} \mathbb{E}_{z \sim p_z} \mathbb{E}_{p_{x,y|z}} \log(f(x, y, z)) - \mathbb{E}_{z \sim p_z} \log \mathbb{E}_{p_{x|z}p_{y|z}} f(x, y, z). \tag{27}$$

Now using the $\eta$-trick for each $z$, followed by the interchangeability principle, we have the final variational form:

$$I(X; Y|Z) = \sup_{f \in \mathscr{F}, \eta} \mathbb{E}_{z \sim p_z} \mathbb{E}_{p_{x,y|z}} f(x, y, z) - \mathbb{E}_{z \sim p_z} \left( e^{-\eta(z)} \mathbb{E}_{p_{x|z}p_{y|z}} e^{f(x,y,z)} + \eta(z) - 1 \right). \tag{28}$$

## D  JIGSAW SELF-SUPERVISION AND MUTUAL INFORMATION

**Jigsaw Task Preprocessing**  For Jigsaw self-supervision task, the encoder needs to learn features from images randomly cropped and tiled into $3 \times 3$ jigsaw puzzles. In order to avoid shortcuts based on matching edges of patches or picking up on color aberration as discussed in Noroozi & Favaro (2016), images are resized to $292 \times 292$, randomly cropped into $255 \times 255$ and jigsawed into $3 \times 3$ or $85 \times 85$ per patch wherein a $64 \times 64$ patch is randomly cropped, avoiding potentially easy edge matching. $255 \times 255$ crops are randomly assigned to gray scales with a probability of ⅔ prior to patch extraction. Each patch RGB channel is normalized to zero mean, unit variance. At test time, the $255 \times 255$ crops are obtained from images center crops.

**ImageNet 10%**  For the Jigsaw experiments, we decided to subsample ImageNet to 10% within *each* class for both training and validation sets (128K training and 5K validation images). The reason is that running on all of ImageNet 1.3 Million images was not practical for reasonable training turnaround on our infrastructure given what we had to train one set of encoder models from the pretext task and one set of classifier models for the target task. Even though we only train classifiers for the target task (as explained later in this section), we still need to forward input features through the Encoder which can still be computationally costly despite only backpropagating through the classifier. This 10% subset of ImageNet was proposed by (Kolesnikov et al., 2019) and can be used for faster training turnaround while still being larger than the Tiny ImageNet dataset (200 classes, 500 training, 50 vaildation, 50 test images per class) often used in publications.

**Jigsaw CE Training** For self-supervision *Jigsaw* pretext task (Noroozi & Favaro, 2016; Kolesnikov et al., 2019) we aim at solving an image jigsaw puzzle by predicting the scrambling permutation $\pi$. Given an image $X$ with $x = \{x_1, \ldots, x_9\}$ 3×3 jigsaw patches, and a permutation $y = \pi_i : [1, 9] \to [1, 9]$, scrambled patches $x_{\pi_i} = \{x_{\pi_i(1)}, \ldots, x_{\pi_i(9)}\}$ generates the puzzle. The Encoder $E$ is based on a ResNet50 and outputs 2048 features for each patch. The classifier $C$ (or "descrambler" in this case) is a simple Neural Network with a linear projection upfront from 2048 to 512 for each patch features that are then stacked (1×(9*512)) and linearly projected to a 512 dimensional subspace, followed by batch normalization, ReLU, dropout, and another linear projection from 512 to 100 (number of our subset of permutations from Noroozi & Favaro (2016)). $E$ and $C$ area trained with minibatch made from 16 permutations of 10 images (160 effective minibatch size). The 10 images and 16 permutations are randomized for each minibatches. For both $E$ and $C$ updates, we use ADAM with ($\beta_1 = 0.9$, $\beta_2 = 0.999$) with a learning rate of $10^{-4}$ that stays fixed for all epochs and no weight decay. We let the model build for a total of 200 epochs with a random seed fixed (for reproducibility). We perform 4 independent builds with 4 different random seeds. We let the model build and just survey the Jigsaw solving accuracy assess the health of training only, not to select and encoder model as performance in pretext task is not a good predictor of performance in the target task (as observed in Kolesnikov et al. (2019). We run for 200 epochs so to be able to turnaround results fast on our infrastructure, not for the goal of early stopping as no overfitting was observed.

**Jigsaw MI training** For MI Jigsaw training, we train $E$ so to maximize MI such that $\max_E I\left(E(x_{\pi(1)}), \ldots, (x_{\pi(9)}); \pi\right)$, which means minimizing the following losses derived from Eq. 8:

$$L_{\text{DV}}(f) = \log \mathbb{E}_{x,y \sim P_x P_y} \left[\exp\left(f(x, y)\right)\right] - \mathbb{E}_{x,y \sim P_{xy}} \left[f(x, y)\right] \qquad \text{(DV-MINE algo.)}$$

$$L_{\eta\text{-DV}}(f, \eta) = e^{-\eta} \mathbb{E}_{x,y \sim P_x P_y} \left[\exp\left(f(x, y)\right)\right] - \mathbb{E}_{x,y \sim P_{xy}} \left[f(x, y)\right] + \eta - 1, \qquad \text{($\eta$-MINE algo.)}$$

where $x$ are scrambled images patches, and $y$ are scrambling permutations. $(x, y) \sim p_x p_y$ are sampled from marginals $p_x$ and $p_y$ ($x$ was not scrambled by permutation $y$) while $(x, y) \sim P_{xy}$ are sampled from the joint distribution ($y$ permutation scrambled $x$). For MI training, we need to define the witness function $f(x, y)$. $x$ is the set of 9 patches feature vectors from $E$ and $y$ is a one-hot embedding of our 100 permutations from Noroozi & Favaro (2016). $x$ and $y$ are first linearly projected in a common dimensionality of 512. Then $x$ goes through a shallow network made of 2 sequential stages each composed of CNN, batch normalization, ReLU, and max pooling. After these 2 stages, we have about a combined 9216 features that we project to 512 dimensional space (same as dimension used for the projection of $y$). $f(x, y)$ is then just the dot product of 512 feature vectors from $x$ and $y$ once normalized, i.e. a cosine distance that can give scores between -1 and 1. We train using minibatches of 10 images and 16 permutations (160 effective minibatch size) as sampling from the joint distribution $p_{xy}$, allowing for 16 unused permutations for sampling from marginal distribution $p_x p_y$. Therefore expectation terms in the losses above for joint and marginal distribution samples are using the same number of samples (160) for each minibatch. Again, we use ADAM for $E$ and $f(x, y)$ with ($\beta_1 = 0.9$, $\beta_2 = 0.999$) with a learning rate of $10^{-4}$ that stays fixed for all epochs and no weight decay. For $\eta$-MINE, we use ADAM for $\eta$ with the same parameters as for $E$ and $f(x, y)$ but with a learning rate of $10^{-2}$ to allow for faster updates of $\eta$. $\eta$ is initialized to 0 for $\eta$-MINE. We run models for 200 epochs of training. A fixed random seed is given to ensure reproducibility. We use 4 seeds to ensure variability of our model builds (same seeds as used for Jigsaw CE). Once again, we observe the training and validation loss as indicators of a training's health and observe no overfitting.

**ImageNet 10% Target Task** For our target classification task, Encoders $E$ from 200 epochs of Jigsaw Task training are frozen and used to train a classifier $C$ for ImageNet 10% with 1000 output classes. $C$ is composed only of a linear projection from 2048 to 1000 classes. For each input image, we resize the images to 232×232 and take a 192×192 center crop before turning it into a 3×3 cells grid where each individual patch (now 64×64) is forwarded through $E$ to get a 2048-dimensional vector for each of the 9 patches. Then, we take the mean of these 9 vectors and pass this 2048-dim mean vector to the classifier $C$ as done in Kolesnikov et al. (2019). Only the classifier $C$ is trained using ADAM with ($\beta_1 = 0.9$, $\beta_2 = 0.999$) with a learning rate of $10^{-4}$ that stays fixed for all epochs and no weight decay. We let $C$ build for a total of 200 epochs with a random seed fixed (for reproducibility); we used 2 seeds for each classification build, for a total of 8 final model configuration (4 seeds for CE and MI Jigsaw training and 2 seeds for Target Task task classification). The reason for 4 and 2 seeds is just so that our infrastructure can handle all the required builds (200 epochs of training each time). We then report average and standard deviation over a set of 8 accuracies (top1,

top5 and top10 in %) on the validation set (5K) in Tab. 3. We used ADAM with these parameters for all classifier builds. Here again random seeds were fixed.

**Model CE**    We built a "baseline" we call mode CE in Tab. 3. The CE model is an attempt at providing a meaningful upper-bound on the performance on our MI trained Encoders by using a Jigsaw CE trained encoder, freezing it and just learning $C$. 200 epochs of training is used, 2 random seeds as well. Despite being better than $\eta$-MINE (and DV-MINE), the difference is only 1.9% absolute for top1 (for $\eta - MINE$). This means that our MI training, and particularly $\eta$-MINE, does a respectable job at learning a useful representation space for the target task.

