# OpenReview forum: "Improved Mutual Information Estimation"
_ICLR.cc/2020/Conference — Reject_

### Official Review · AnonReviewer1 · 2019-10-22
**Official Blind Review #1**

**Rating:** 1

**Review:**

This paper develops variations on the Donsker-Varadhan (DV) lower bound on KL divergence which yields a lower bound on mutual information as a special case.  The paper is primarily theoretical with an emphasis on the case where the witness function f is drawn from an RKHS (a support vector machine).  They discuss, and present experimental results where the feature map of the RKHS is computed by a neural network, although the theoretical results largely do not apply to optimization of the neural network.

I have two complaints.  First, the authors ignore fundamental limitations on the measurement of mutual information from finite samples.  See "Fundamental Limitations on the Measurement of Mutual Information" by McAllester and Stratos.  This paper makes the intuitively obvious observation that I(X,Y) <= H(X) and one cannot give meaningful lower bound on H(X) larger than about 2 log N when drawing only N samples --- the best test one can use is the birthday paradox and a non-repetitive sample only guarantees that the entropy is larger than about 2 log N.  This is obvious for discrete distributions but holds also for continuous distributions --- from a finite sample one cannot even tell if the distribution is continuous or discrete.   So meaningful lower bounds for "high dimensional data" are simply not possible for feasible samples.  Given this fact, the emphasis needs to be on experimental results.

The experimental results in this paper are extremely weak. They should be compared to those in "Learning Representations by Maximizing Mutual Information Across Views" by Philip Bachman, R Devon Hjelm, William Buchwalter


Response to the author response:

This was written earlier but there was a mishap when I attempted to submit it and it was not actually submitted until comments were no longer available to the authors so I am putting this in the review.

Bounding the ratio of the densities already bounds the mutual information.  In order for the actual mutual information to be large (hundreds of bits) the log density ratios must actually be extreme.  We do believe, presumably, that mutual information in video data can be hundreds of bits.  So I am still not convinced that lower bounds on large quantities of mutual information are meaningful.

Also, your bound, like the DV bound, involves an empirical estimator $\frac{1}{N} \sum_{i=1}^N\; e^{f(x)}$ of an expectation $E_x e^{f(x)}$.  The true expectation of an exponential seems likely to be dominated by rare extremes of f(x) which contribute exponentially to the expectation.  I see no reason to believe that the empirical estimate is meaningful in a real application such as vision.

Regarding the experiments, I do not have much interest in experiments on synthetic Gaussians.  The most meaningful experiments for me are the pre-training results for CIFAR.  But you seem to be comparing yourself to your own implementations of (weak?) baselines rather than performance numbers reported by, for example, Hjelm et al.  or van den Oord et al. (CPC).



**Experience Assessment:**

I have published one or two papers in this area.

**Review Assessment: Checking Correctness Of Derivations And Theory:**

I assessed the sensibility of the derivations and theory.

**Review Assessment: Checking Correctness Of Experiments:**

I assessed the sensibility of the experiments.

**Review Assessment: Thoroughness In Paper Reading:**

I made a quick assessment of this paper.

---

> ### Author Response · Authors · 2019-11-14
> **Thank you for your comment, we respectfully disagree**
>
> We thank the reviewer for their review. We respectfully disagree with them on both complaints they make.
>
> On the theoretical part, the reviewer did not check the theory presented in the work but rather questioned the problem studied in this paper.
>
> Thank you for the reference, while this reference gives a pessimistic result in the sense of the worst case analysis. In general, under mild assumptions on the mutual information and the function class, one can get finite bounds for estimating mutual information from samples.
>
> Under the assumption that the mutual information is finite and that the ratio of the densities is bounded from above and below and some mild assumptions on the function class (that are satisfied in the case of an RKHS), Nguyen, Wainwright, and Jordan showed that the mutual information can be estimated from finite samples with an error O(1/sqrt(N)), N being the number of samples. (Please see page 11 http://www2.stat.duke.edu/~xn3/Papers/Mutin_Tech_Final_v2.pdf ). Those results translate also to our case when the estimation is done in an RKHS.
>
> On the experimental side, can the reviewer elaborate on which part is weak in our experiment? In fact, we disagree that the experiments are weak, since we showed that on multiple applications: in GAN stabilization and two applications in self-supervised learning we showed that our method outperforms other estimators. The estimator in the paper  suggested by the reviewer is based on NCE  with an emphasis on 1) architecture choice 2) local mutual information and are tailored towards the specific application of learning representations which is not the only focus of our work. We compared NCE to our method in the synthetic case and in the self-supervised case we limited our comparison to DV-type bounds rather than comparing to other bounds.

---

### Official Review · AnonReviewer3 · 2019-10-23
**Official Blind Review #3**

**Rating:** 3

**Review:**

The paper presents to use \eta-trick for log(.) in Donsker-Varadhan representation of the KL divergence. The paper discusses its dual form in RKHS and claims it is better than the dual form of f-divergence (while I don't think it's a correct claim). Nevertheless, in experiments, we see that it outperforms the original neural estimation of mutual information.

[Overall]
Pros:
1. The idea of avoiding the log-sum-exp computation in the DV representation of KL is good. One of the main reasons is to get rid of biased estimation. This idea may not be too novel, but definitely is useful.

Cons:
1. I don't agree with some claims in the paper. Nevertheless, these claims are some of the main stories supporting the paper.
2. The presentation of the paper should be improved. Including the presentation flow between sections, and also misleading part in experiments.
3. There are TOO MANY typos in equations.

[Cons In Details]
<The role of discussing the dual formulation.>
The paper spends huge paragraphs discussing the role of the dual formulation. And it also introduces \eta-DV-Fisher and \eta-DV-Sobolev, which can be seen as extensions of the proposed method.
Nevertheless, the author doesn't present an evaluation using the dual form. It's a pity that this part is missing. Having Section 3 makes the paper contain several sporadic arguments unrelating to the research questions. Re-organize the paragraphs/ presentation is suggested.
Another example is introducing perturbed loss function into the proposed loss function to make it strictly convex. This paragraph is misleading and can be moved entirely to the Appendix.

<Claim on the Proposed Method is Better than f-divergence>
The author emphasizes (in multiple places) that the f-divergence biases the estimate. And it exemplifies from eq. (15) to eq. (16). Nevertheless, in eq. (16), when r* is optimum, there should be no second term. The author's claim is based on the comparisons between eq. (13) and eq. (15) when assuming only the MMD term reaches zero. The statement may not be fair.
<Section 4>
Similar to Section 3, this section is cluttered. I don't get the reason why the author specifically come up with a new section comparing only to one mutual information estimation approach (a-MINE). Another irrelevant part of the research question is point 4 under page 7. Why discussing the extension of the a-MINE to conditional MI estimation?
Some Typos: In equation (21) and (22), \eta is missing. In Algorithm 1, there are too many typos such as missing \theta under f, entirely wrong equation for the Output.

<MI Estimation Experiment>
Can the author discuss the standard deviation for various MI estimation approaches? The large standard deviation for MINE seems unusual.

<Self-Supervised Experiment>
The author mentioned that the network considering only the first two convolutional layers, followed by a linear classifier, leads to the best performance. Is there no other layer in between? Also, in Figure 4 (a) and (c), is the purple-color layer means the final classification layer? It is a bit confusing.

<Appendix>
I understand most of the people would not read the Appendix, but I do. Missing brackets, grammatic errors,  missing integrals, wrong notations, missing punctuation marks, ill-structured presentations, etc., are the problems in the Appendix. I would greatly appreciate the author also spend some time in the Appendix.

[Summary]
I would vote for a score with 4 or 5 to this paper.
Regarding there're only 3/6, I'm proposing a score of 6 now. But I look forward to the authors' response and then addressing the problems that I identified. I feel the paper should be a strong paper after a good amount of revision.

**Experience Assessment:**

I have published in this field for several years.

**Review Assessment: Checking Correctness Of Derivations And Theory:**

I assessed the sensibility of the derivations and theory.

**Review Assessment: Checking Correctness Of Experiments:**

I carefully checked the experiments.

**Review Assessment: Thoroughness In Paper Reading:**

I read the paper thoroughly.

---

> ### Author Response · Authors · 2019-11-14
> **Thank you for your careful review, we implemented your suggestions and added clarifications**
>
> Thank you for your careful review and for your suggestions , we improved the manuscript in the revision. We address in what follows your main concerns.
>
> *Q1: The role of the dual formulation*
>
> Since the dual and the primal formulation are equivalent in RKHS, the goal of this section was only to show the equivalence of the primal to ratio estimation and the role of eta in enforcing a normalizing constraint on the ratio. We have made this section easier to follow per your suggestion. Please check the revision.
>
> *Q2: Comparaison to f-divergence*
>
> We make the following clarification our claims are fair and rigorous on the biased nature of f-divergence.
>
> We are not assuming the mmd is zero to make this comparison.
>
> Given $r^*$ , by duality we have $w^*=\int \Phi(x)p(x)- \int r^*(x)\Phi(x)q(x)$, and hence in f-divergence  the loss is (noting that mmd is the norm of $w^*$):
>
> $$L_{f-div}(w^*) = \int r^* \log r^* q + (1-\int r^* q ) + \frac{2}{\lambda}||w^*||^2,$$
>
> A similar expression holds for $\eta$ DV:
> $$L_{\eta-DV}(w^*,\eta^*)= \int r^* \log r^* q + \frac{2}{\lambda}||w^*||^2, $$
>
> To get an estimate of $KL$ we are following Nguyen et al 2008 in discarding the regularization terms for the losses above (see Eq 44 page 15 here http://www2.stat.duke.edu/~xn3/Papers/Mutin_Tech_Final_v2.pdf ). This is standard in Thikonov regularized problems we should just look at the loss and not the regularization term to get an estimate of the loss function of interest.  Hence we see that the estimate of $\eta $ DV is tighter than the one of $f$- div.
>
> *Q3: Section 4 and A-mine*
>
> The main rationale behind this section was to compare our approach to the closest previous work: the a-MINE approach (Poole et al). We believe this discussion is scientifically important : we give an improved Donsker Varadhan formulation for the special case of mutual information and use the interchangeability principle to get rigorously an unbiased variational formulation for this.
>
> Please note a-MINE,  instead of having a single lagrangian (eta), it uses a parameterized  neural network of Lagrangians (eta(x)). We showed that this is not needed for the purpose of mutual information estimation, and a single Lagrangian is enough. To be fair with respect to the a-MINE approach, we noted that their approach can be useful in the conditional mutual information case.
>
> *Q4: Standard deviation in MI estimation experiments*
>
> For 2-D Gaussians (Fig, 3(a)), the standard deviation of MI estimates for MINE estimator is in fact comparable to the standard deviation of other methods (e.g., a-MINE or f-MINE). Moreover, please note the value range in all the plots: the apparent large size of bands in 2-D Gaussians is due to the small range (0-0.5), while similar size bands appear much smaller in the 20-D Gaussians, where the range is (1-4). To avoid any confusion, we will clarify this in the paper.
>
>
> *Q5: Self-Supervised Experiment in Fig. 4*
> You are correct, we kept the architecture very simple and the best performing network had only two convolutional layers, followed by the linear classification layer. In Figure 4(c), the final classification layer (which generates logits over the classes) is the grey-colored box. In Figure 4(a), the final classification layer is not really shown since it depends on which part of the network is used for pre-training, but in all the cases the linear layer would transform the data of different size into the logits over the same number classes.
>
>
> *Q6: Appendix *
>
> Thank you for your careful reading. We have updated the appendix and made sure all comments and suggestions on typos were implemented. We added a table of content and transitions between sections to make the appendix more coherent for the reader.

---

> > ### Comment · AnonReviewer3 · 2019-11-15
> > **Thank you for the response**
> >
> > We thank the author for the response. Although some of the concerns are addressed, the main two concerns are not addressed:
> >
> > 1. The presentation flow requires a significant effort to improve.
> >
> > 2. The role of the dual function is still not clear.  It may also be the case that the presentation makes the role of the dual function hard to follow.
> >
> > Therefore, I would downgrade my score to 3.

---

> > > ### Author Response · Authors · 2019-11-15
> > > **Thank you , let us try to clarify this point**
> > >
> > > Thank you for your answer, we are sorry our revision did not make this point clearer.
> > >
> > > Could you please tell us what is still unclear in the role of the dual?
> > >
> > > Let us try to clarify  :
> > >
> > > Understanding the role of $\eta$ not only in un-biasing the primal problem is interesting and important.
> > >
> > > The critic function of the primal is estimating a ratio $f= Log (p/q)$.  Hence the estimation of the critic of the primal is implicitly a likelihood ratio estimation.
> > >
> > > What we show is that indeed the DV formulation written equivalently with the $\eta-$ trick ($\eta-$ DV) has a dual that boils down explicitly to a likelihood ratio estimation. And moreover this ratio is guaranteed to be  normalized thanks to the role of $\eta$ as Lagrangian that imposes this constraint. Hence $\eta-$ DV allows a better estimation of the  likelihood ratio (the dual variable  ) and subsequently the critic  (the primal variable) is also better estimated.
> > >
> > > The dual here is to understand how $\eta$ helps in getting better estimate of the critic and subsequently the value of the lower bound of the KL divergence.
> > >
> > >  If this explanation helps please let us know , we will revise the paper to include it and further improve the flow of the paper.

---

### Official Review · AnonReviewer2 · 2019-10-23
**Official Blind Review #2**

**Rating:** 6

**Review:**

The paper propose a variational bound on the Mutual Information and showed how it can be used to improve the estimation of mutual information.

I am afraid I cannot judge of the quality and correctness of the method introduced by the authors. I am not familiar with the subject and will be a poor judge of the quality of the paper. Nevertheless I find that the presentation of the paper could be improved.

For instance,  while I enjoyed Fig. 3 that showed the performances of different estimators of the entropy, i add the zoom out on a big screen to be able to see anything at all! Clearly, one the most important figure of the paper will be unreadable when printed! It is also not entirely clear how this figure supports the claim of superiority of the proposed method.

The only comment I may have is that it would be interesting --since the authors want to apply their bound to the case of neural networks-- to compare with, the rigorous estimation of the entropy of NN with random weights in Gabrie et al, NeurIPS 2018, Fig. 2 . It would be a much challenging task, albeit a synthetic one, than the Gaussian dataset one presented in Fig. 3.






**Experience Assessment:**

I do not know much about this area.

**Review Assessment: Checking Correctness Of Derivations And Theory:**

I did not assess the derivations or theory.

**Review Assessment: Checking Correctness Of Experiments:**

I did not assess the experiments.

**Review Assessment: Thoroughness In Paper Reading:**

I made a quick assessment of this paper.

---

> ### Author Response · Authors · 2019-11-14
> **Thank you for your comments**
>
> We thank the reviewer for their  comments  and for acknowledging that they are not familiar with the area of our submission and that they did not assess the theory and the experiments in the paper.
>
> We improved the presentation and readability of Figure 3 as suggested by the reviewer.
>
> Thank you for your suggestion, this is indeed an interesting setting to test the estimator, but we presented more challenging experiments  in learning representations in the self-supervised setting using neural mutual information estimators.

---

### Official Review · AnonReviewer4 · 2019-11-15
**Official Blind Review #4**

**Rating:** 3

**Review:**



The paper contains one main contribution, the unbiased version of MINE obtained using the eta-trick, however a lot of theory is presented and not always very clearly. While some results are interesting (like the constrained ratio estimation interpretation of the dual) some parts are unclear and of lesser relevance. The section on “What Do Neural Estimators of LK or MI Learn” is a collection of different remarks without much coherence, some of which are imprecisely stated. The comparison with the estimator from Pool et al. (2019) could also be much simplified, in particular I would review the list of remarks below theorem 2.

Another weakness of the paper is that two important aspects in assessing the quality of an estimator are overlooked:  the variance of the estimator and the performance on finite data. Bias is not the only property which matters, both the variance and the dependence on the number of samples should be assessed experimentally, especially when no discussion or theoretical results are provided.

I liked to see experiments performed on different tasks and datasets but overall that section could be significantly improved.

The experiment comparing different MI estimators on synthetic Gaussian data is interesting. The plots are however difficult to read, it would be good to make them larger or split the results in different panels. For the 2D and the 20D case, the results reported in the MINE paper are much closer to the true MI than what is reported here, could the authors explain this difference? It would also be good to see some experiments done with a higher ground truth MI, 3.5 is not a lot for higher dimensional cases.

Concerning the Deep InfoMax experiment, we see some improvements when using the proposed MI estimator with Deep InfoMax, however I doubt that this task is an ideal test case for an MI estimator since it has been shown (On Mutual Information Maximization for Representation Learning, Tschannen et al.) that the performance on downstream supervised tasks often does not clearly depend on the quality of MI estimation.


Some additional points:

Assuming a finite dimensional feature map in section 3 actually is a loss of generality.

The proof of lemma 3 is  hard to follow with some notations used without being properly introduced or changed in unexpected ways (eta switches places in f_b(w, eta), what does q do here?, what does PSD mean?).

The proof of theorem 1 is again unclear. It starts with a typo in the second line (E_y~Q should be replaced by an integral). A couple of lines below a function alpha is introduced without further explanation.

Typos:

There is one x missing in the proof of Lemma 1 Appendix A.







**Experience Assessment:**

I have published one or two papers in this area.

**Review Assessment: Checking Correctness Of Derivations And Theory:**

I assessed the sensibility of the derivations and theory.

**Review Assessment: Checking Correctness Of Experiments:**

I assessed the sensibility of the experiments.

**Review Assessment: Thoroughness In Paper Reading:**

I read the paper thoroughly.

---

> ### Author Response · Authors · 2019-11-15
> **Thank you for your review!**
>
> Thank you for your review! Our revision greatly improved the presentation (figures etc ..)
>
> "Another weakness of the paper is that two important aspects in assessing the quality of an estimator are overlooked:  the variance of the estimator and the performance on finite data. Bias is not the only property which matters, both the variance and the dependence on the number of samples should be assessed experimentally, especially when no discussion or theoretical results are provided. "
>
> The dependence on samples being a kernel method and assuming that the ratio belongs to the RKHS and that the mutual information is finite,  is  with an error bound of O(1/sqrt{N}) . Please see our response to Reviewer #1. This will follow from similar analysis to Nguyen et al 2008. We omitted this to not clutter the paper.
>
>
> "The experiment comparing different MI estimators on synthetic Gaussian data is interesting. The plots are however difficult to read, it would be good to make them larger or split the results in different panels. For the 2D and the 20D case, the results reported in the MINE paper are much closer to the true MI than what is reported here, could the authors explain this difference? It would also be good to see some experiments done with a higher ground truth MI, 3.5 is not a lot for higher dimensional cases."
>
> Unfortunately in the mine paper they considered high dimensional gaussians with constant cross correlation which makes the estimation almost like the one dimensional case (estimating the diagonal and this scalar cross-correlation) . We used the difficult setting of  general covariances , that show that the problem is indeed more difficult as dimension grows.
>
> "Concerning the Deep InfoMax experiment, we see some improvements when using the proposed MI estimator with Deep InfoMax, however I doubt that this task is an ideal test case for an MI estimator since it has been shown (On Mutual Information Maximization for Representation Learning, Tschannen et al.) that the performance on downstream supervised tasks often does not clearly depend on the quality of MI estimation. "
>
> Application in self-supervised learning in deep-infomax  are not here indeed to show the strength of the estimator that was proven in the synthetic case, but to show its versatility of the estimator  and the help of the unbiased nature of our  estimator in subsequent tasks.
>
> Other points:
>
> "Assuming a finite dimensional feature map in section 3 actually is a loss of generality.  "
>
> This is not the case , we just used this for the ease of presentation , if we use a infinite dimensional feature map the kernel trick applies and the dual is similar to SVM duals that can be solved to give estimate of mutual information, we wanted to avoid the cumbersome   computation of dual problems with kernels.
>
> "The proof of lemma 3 is  hard to follow with some notations used without being properly introduced or changed in unexpected ways (eta switches places in f_b(w, eta), what does q do here?, what does PSD mean?). "
>
> We fixed those typos. PSD is positive semi definite.
>
> "The proof of theorem 1 is again unclear. It starts with a typo in the second line (E_y~Q should be replaced by an integral). A couple of lines below a function alpha is introduced without further explanation."
>
> Thank you we fixed this typo. alpha is the lagrangian, we added an explanation.

---

### Author Response · Authors · 2019-11-14
**General Response**

We thank the reviewers for their comments. We made a revision of the paper to incorporate their feedback. We uploaded a new pdf on open review with the following revisions:

1- Per Reviewer #3 suggestion we moved the part on discussion of other regularizers such as fisher and Sobolev to the appendix.

2- Per Reviewer #3 suggestion we fixed typos in the appendix and made it easier to follow.

3- We improved the presentation of Figure 3 per Reviewer #2 suggestion

Please take a look on the updated manuscript. Thank you !

---

### Decision · Program_Chairs · 2019-12-19

**Decision:**

Reject

**Comment:**

This paper centers on an unbiased variant of the Mutual Information Neural Estimation (procedure), using the so-called "eta trick" applied to the Donsker-Varadhan lower bound on the KL divergence. The paper's contribution is mainly theoretical though experiments are presented on synthetic Gaussian-distributed data as well as CIFAR10 and STL10 classification experiments (from learned representations).

R1's criticism of the theoretical contributions centers on fundamental limitations on finite sample estimation of the MI, contending that the bounds simply aren't meaningful in high-dimensional settings, and that the empirical work centers on synthetic data and self-generated baselines rather than comparisons to reported numbers in the literature; they were unswayed by the author response, which contended that these criticisms were based on pessimistic worst-case analysis and that "mild assumptions on the mutual information and function class" could render better finite-sample bounds. Some of R3's concerns were addressed by the author rebuttal and associated updates, but remained critical of the presentation, in particular regarding the dual function, and downgraded their score.

Because R2 disclosed that they were outside of their area of strong expertise, a 4th reviewer was sought (by this stage, the paper was the revised version). Concerns about clarity persisted, with R4 remarking that a section was "a collection of different remarks without much coherence, some of which are imprecisely stated". R4 felt variance and sample complexity should be dealt with experimentally, though this was not directly addressed in the author response. R4 also remarked that the plots were difficult to read and questioned the utility of supervised representation learning benchmarks at assessing the quality of MI estimation, given recent evidence in the literature.

The theoretical contributions of this submission are slightly outside the bounds of my own expertise, but consensus among three expert reviewers appears to be that the clarity of exposition leaves much to be desired, and I concur with their assessment that the empirical investigation is insufficiently rigorous and does not draw clear comparisons to existing work in this area. I therefore recommend rejection.